# Patch-Based Diffusion Models Beat Whole-Image Models for Mismatched Distribution Inverse Problems

## Abstract

Diffusion models have achieved excellent success in solving inverse problems due to their ability to learn strong image priors, but existing approaches require a large training dataset of images that should come from the same distribution as the test dataset. When the training and test distributions are mismatched, artifacts and hallucinations can occur in reconstructed images due to the incorrect priors. In this work, we systematically study out of distribution (OOD) problems where a known training distribution is first provided. We first study the setting where only a single measurement obtained from the unknown test distribution is available. Next we study the setting where a very small sample of data belonging to the test distribution is available, and our goal is still to reconstruct an image from a measurement that came from the test distribution. In both settings, we use a patch-based diffusion prior that learns the image distribution solely from patches. Furthermore, in the first setting, we include a self-supervised loss that helps the network output maintain consistency with the measurement. Extensive experiments show that in both settings, the patch-based method can obtain high quality image reconstructions that can outperform whole-image models and can compete with methods that have access to large in-distribution training datasets. Furthermore, we show how whole-image models are prone to memorization and overfitting, leading to artifacts in the reconstructions, while a patch-based model can resolve these issues.

## 1 Introduction

In image processing, inverse problems are of paramount importance and consist of reconstructing a latent image $\boldsymbol{x}$ from a measurement $\boldsymbol{y} = \mathcal{A}(\boldsymbol{x}) + \boldsymbol{\varepsilon}$. Here, $\mathcal{A}$ represents a forward operator and $\boldsymbol{\varepsilon}$ represents random unknown noise. By Bayes' rule, $\log p(\boldsymbol{x}|\boldsymbol{y})$ is proportional to $\log p(\boldsymbol{x}) + \log p(\boldsymbol{y}|\boldsymbol{x})$, so obtaining a good prior $p(\boldsymbol{x})$ is crucial for recovering $\boldsymbol{x}$ when $\boldsymbol{y}$ contains far less information than $\boldsymbol{x}$. Diffusion models obtain state-of-the-art results for learning a strong prior and sampling from it, so similarly competitive results can be obtained when using them to solve inverse problems (Chung et al., 2022a; 2023a; Song et al., 2024; Wang et al., 2022; Kawar et al., 2021; Li et al., 2023a).

However, training diffusion models well requires vast amounts of clean training data (Song et al., 2021; Ho et al., 2020), which is infeasible to collect in many applications such as medical imaging (Chung et al., 2022b; Song et al., 2022; Jalal et al., 2021), black hole imaging (Feng et al., 2023; 2024), and phase retrieval (Li et al., 2023a; Wu et al., 2019). In particular, no ground truth images are known for very challenging inverse problems such as black hole imaging (Feng et al., 2023) and Fresnel phase retrieval (Gureyev et al., 2004), so one must obtain a reconstruction from only a single measurement $\boldsymbol{y}$. In other applications such as dynamic CT reconstruction (Reed et al., 2021) and single photo emission CT (Li et al., 2023b), obtaining a high quality measurement that can lead to a reconstruction that closely approximates the ground truth can be slow or potentially harmful to the patient, so only a very small dataset of clean images are available. Thus, in this paper we consider two settings: the *single measurement* setting in which we are given one measurement $\boldsymbol{y}$ whose corresponding $\boldsymbol{x}$ belongs to a different distribution from the training dataset, and the *small dataset* setting in which we are only given a small number of samples $\boldsymbol{x}$ that belong to the same distribution as the test dataset.

In recent years, some works have aimed to address these problems by demonstrating that diffusion models have a stronger generalization ability than other deep learning methods (Jalal et al., 2021), so slight distribution mismatches between the training data and test data may not significantly degrade the reconstructed image quality. However, in cases of particularly compressed or noisy measurements, as well as when the test data is severely out of distribution (OOD) with a significant domain shift, an improper choice of training data leads to an incorrect prior that causes substantial image degradation and hallucinations (Feng et al., 2023; Barbano et al., 2023). To address these challenges in the single measurement case, recent works use each measurement $\boldsymbol{y}$ to adjust the weights of a diffusion network at reconstruction time Barbano et al. (2023); Chung & Ye (2024), aiming to shift the underlying prior learned by the network toward the appropriate prior for the latent image in the test case. However, as the networks have huge numbers of weights, an intricate and parameter-sensitive refining process of the network is required during reconstruction to avoid overfitting to the measurement. Furthermore, there is still a substantial gap in performance between methods using an OOD prior and methods using an in-distribution prior. Finally, these methods have only been tested in medical imaging applications (Barbano et al., 2023; Chung & Ye, 2024). On the other hand, in the small dataset case, various methods (Moon et al., 2022; Zhang et al., 2024) have been devised to fine-tune a diffusion model on an OOD dataset, but these methods still require several hundred images and have not used the fine-tuned network to solve inverse problems.

Patch-based diffusion models have shown success both for image generation (Wang et al., 2023; Ding et al., 2023) and for inverse problem solving (Hu et al., 2024). In particular, the method of Hu et al. (2024) involves training networks that take in only patches of images at training and reconstruction time, learning priors of the entire images from only priors of patches. In cases of limited data, Hu et al. (2024) shows that patch-based diffusion models outperform whole image models for solving certain inverse problems. These works motivate our key insight that patch-based diffusion priors potentially obtain stronger generalizability than whole-image diffusion priors for both the single measurement setting and the small dataset setting due to a severe lack of data. Inspired by this, we propose to utilize patch-based diffusion models to tackle the challenges arising from mismatched distributions and lack of data in a unified way. We first develop a method to take a network trained on patches of a mismatched distribution and adjust it on the fly in the single measurement setting. We also show how in the small dataset setting, fine-tuning a patch-based network results in a much better prior than fine-tuning a whole-image network, leading to higher quality reconstructed images.

In summary, our contributions are as follows:

- We integrate the patch-based diffusion model framework with the deep image prior (DIP) framework to correct for mismatched distributions in the single measurement setting. Experimentally, we find this approach beats using whole-image models in terms of quantitative metrics and visual image quality in image reconstruction tasks, as well as achieving competitive results with methods using in-distribution diffusion models.

- We show that in the small dataset setting, fine-tuning patch-based diffusion models is much more robust than whole-image models and very little data is required to obtain a reasonable prior for solving inverse problems.

- We demonstrate experimentally that when fine-tuning on very small datasets, whole image diffusion models are prone to overfitting and memorization, which severely degrades reconstructed images, while patch-based models are much less sensitive to this problem.

## 2 BACKGROUND AND RELATED WORK

**Diffusion models and inverse problems.** In a general framework, diffusion models involve the forward stochastic differential equation (SDE)

$$\mathrm{d}\boldsymbol{x}_t = -\frac{\beta(t)}{2}\,\boldsymbol{x}_t\,\mathrm{d}t + \sqrt{\beta(t)}\,\mathrm{d}\boldsymbol{w}_t, \tag{1}$$

where $t \in [0, T]$, $\boldsymbol{x}(t) \in \mathbb{R}^d$, and $\beta(t)$ is the noise variance schedule of the random process $\mathrm{d}\boldsymbol{w}(t)$. This process adds noise to a clean image and ends with an image indistinguishable from Gaussian noise (Song et al., 2021). Thus, the distribution of $\boldsymbol{x}(0)$ is the data distribution and the distribution

of $\boldsymbol{x}(T)$ is (approximately) a standard Gaussian. Then the reverse SDE has the form (Anderson, 1982):

$$\mathrm{d}\boldsymbol{x}_t = \left(-\frac{\beta(t)}{2}\boldsymbol{x}_t - \beta(t)\nabla_{\boldsymbol{x}_t}\log p_t(\boldsymbol{x}_t)\right)\mathrm{d}t + \sqrt{\beta(t)}\,\mathrm{d}\bar{\boldsymbol{w}}_t. \tag{2}$$

Score-based diffusion models involve training a neural network to learn the score function $\nabla_{\boldsymbol{x}_t}\log p_t(\boldsymbol{x}_t)$, from which one can start with noise and run the reverse SDE to obtain samples from the learned data distribution.

When solving inverse problems, it is necessary to instead sample from $p(\boldsymbol{x}_t|\boldsymbol{y})$, so the reverse SDE becomes

$$\mathrm{d}\boldsymbol{x}_t = \left(-\frac{\beta(t)}{2}\boldsymbol{x}_t - \beta(t)\nabla_{\boldsymbol{x}_t}\log p_t(\boldsymbol{x}_t|\boldsymbol{y})\right)\mathrm{d}t + \sqrt{\beta(t)}\,\mathrm{d}\bar{\boldsymbol{w}}_t. \tag{3}$$

Unfortunately, the term $\log p_t(\boldsymbol{x}_t|\boldsymbol{y})$ seems difficult to compute from the unconditional score $\nabla_{\boldsymbol{x}_t}\log p_t(\boldsymbol{x}_t)$ alone. Liu et al. (2023), Chung et al. (2023b), and Ozdenizci & Legenstein (2023) among others proposed directly learning this conditional score $\nabla_{\boldsymbol{x}_t}\log p_t(\boldsymbol{x}_t|\boldsymbol{y})$ instead. However, this process requires paired data $(\boldsymbol{x}, \boldsymbol{y})$ between the image domain and measurement domain for training, instead of just clean image data. Furthermore, the learned conditional score function is suitable only for the particular inverse problem for which it was trained, limiting its flexibility.

For greater generalizability, it is desirable to apply the unconditional score $\nabla_{\boldsymbol{x}_t}\log p_t(\boldsymbol{x}_t)$ to be able to solve a wide variety of inverse problems. Thus, many works have been proposed to approximate the conditional score in terms of the unconditional one (Wang et al., 2022; Chung et al., 2023a; 2024; Kawar et al., 2022). Notably, Peng et al. (2024) unified various diffusion inverse solvers (DIS) into two categories: the first consists of direct approximations to $p_t(\boldsymbol{y}|\boldsymbol{x}_t)$, and the second consists of first approximating $\mathbb{E}[\boldsymbol{x}_0|\boldsymbol{x}_t, \boldsymbol{y}]$ (typically through an optimization problem balancing the prior and measurement) and then applying Tweedie's formula (Efron, 2011) to obtain

$$\nabla\log p_t(\boldsymbol{x}_t|\boldsymbol{y}) = \frac{\mathbb{E}[\boldsymbol{x}_0|\boldsymbol{x}_t, \boldsymbol{y}] - \boldsymbol{x}_t}{\sigma_t^2}, \tag{4}$$

where $\sigma_t$ is the noise level of $\boldsymbol{x}_t$. All of these methods require a large quantity of clean training data that should come from the distribution $p(\boldsymbol{x})$ whose score is to be learned, which may not be available in practice.

**Methods without clean training data.** When no in-distribution data is available, one approach is to use traditional methods that do not require any training data, such as total variation (TV) (Li et al., 2019) or wavelet transform (Daubechies, 1992) regularizers that encourage image sparsity. More recently, plug and play (PnP) methods have risen in popularity (Sun et al., 2021; Sreehari et al., 2016; Hong et al., 2020; 2024b); these methods use a denoiser to solve general inverse problems. Although these methods often use a trained denoiser, Ryu et al. (2019) found that using an off-the-shelf denoiser such as block matching 3D (Dabov et al., 2006) can yield competitive results. Nevertheless, with the rise of deep learning in image processing applications, methods that harness the power of these tools may be desirable.

The deep image prior (DIP) is an extensively studied self-supervised method that is popular when no training data is available and reconstruction from a single measurement $\boldsymbol{y}$ is desired. The method consists of training a network $f_{\boldsymbol{\theta}}$ using the loss function

$$L(\boldsymbol{\theta}) = \|\boldsymbol{y} - \mathcal{A}(f_{\boldsymbol{\theta}}(\boldsymbol{z}))\|_2^2, \quad \boldsymbol{z} \sim \mathcal{N}(0, \boldsymbol{I}), \tag{5}$$

so that $f_{\boldsymbol{\theta}}(\boldsymbol{z})$ produces the reconstruction. Although the neural network acts as an implicit regularizer whose output tends to lie in the manifold of clean images, DIP is prone to overfitting (Ulyanov et al., 2020). Various methods have been proposed involving early stopping, regularization, and network initialization (Liu et al., 2018; Jo et al., 2021; Barbano et al., 2022). Nevertheless, the method is very sensitive to parameter selection and implementation and can take a long time to train (Jo et al., 2021).

Most DIS methods learn a prior from a large collection of clean in-distribution training images, but recently Barbano et al. (2023) and Chung & Ye (2024) proposed self-supervised diffusion model methods that are based off the DIP framework. These methods involve alternating between the usual reverse diffusion update step to gradually denoise the image and a network refining step in which the score network parameters are updated via the loss function

$$L(\boldsymbol{\theta}) = \|\boldsymbol{y} - \mathcal{A}(\mathrm{CG}(\hat{\boldsymbol{x}}_{0|t}(\boldsymbol{x}_t; \boldsymbol{\theta})))\|_2^2 \tag{6}$$

where conjugate gradient (CG) descent is used to enforce data fidelity. This CG step consists of solving an optimization of the form

$$\arg \min_{\boldsymbol{x}} \frac{\gamma}{2} \|\boldsymbol{y} - \mathcal{A}(\boldsymbol{x})\|_2^2 + \frac{1}{2} \|\boldsymbol{x} - \hat{\boldsymbol{x}}_{0|t}\|_2^2, \tag{7}$$

where $\gamma$ is a tradeoff parameter controlling the strength of the prior versus the measurement. Crucially, these methods introduce an additional LoRA module (Hu et al., 2021) to the network and the original network parameters are frozen when backpropagating the loss, which helps to avoid overfitting the whole-image model. Nevertheless, many technical tricks are required (Chung & Ye, 2024) involving noisy initializations and early stopping to obtain good results and avoid artifacts. Our patch-based model avoids this overfitting issue.

**Diffusion model fine-tuning.** In the small dataset setting, various fine-tuning methods exist to shift the underlying prior learned by a score network away from a mismatched distribution and toward a target distribution. Given a pretrained diffusion network on a mismatched distribution, Moon et al. (2022), Zhang et al. (2024), and Zhu et al. (2024) among others have studied ways to fine-tune the network to the desired dataset. These methods generally involve freezing certain layers of the original network, appending extra modules that contain relatively few weights, or modifying the loss function to capture details that differ greatly between distributions. However, these methods usually still require thousands of images from the desired distribution and focus on image generation. When solving inverse problems, the reconstructed image should be consistent with the measurement $\boldsymbol{y}$, reducing the number of degrees of the freedom for the image compared to generation, so with proper fine-tuning the data requirement should be lower.

## 3 METHODS

### 3.1 PATCH-BASED PRIOR

We adapt the patch-based diffusion model framework of Hu et al. (2024). We first zero pad the image by an amount $P$ on each side and model the resulting image $\boldsymbol{x}$. When choosing the $i$th patch offset tuple $(o_1, o_2) \in \{0, \dots, P-1\}^2$ in Figure 1, $\boldsymbol{x}$ is partitioned into many square patches and one bordering region consisting of all zeros. Since $k = N/P$ patches are needed in one direction to perfectly cover the image, a model for the data distribution takes the form

$$p(\boldsymbol{x}) = \prod_{i=1}^{P^2} p_{i,B}(\boldsymbol{x}_{i,B}) \prod_{r=1}^{(k+1)^2} p_{i,r}(\boldsymbol{x}_{i,r})/Z, \tag{8}$$

where $\boldsymbol{x}_{i,B}$ represents the bordering region of $\boldsymbol{x}$ that depends on the specific value of $i$, $p_{i,B}$ is the probability distribution of that region, $\boldsymbol{x}_{i,r}$ is the $r$th $P \times P$ patch when using the partitioning scheme corresponding to the value of $i$, $p_{i,r}$ is the probability distribution of that region, and $Z$ is a normalizing factor. This model allows for a variety of possible tilings of the image, eliminating boundary artifacts that would occur if only one tiling was used.

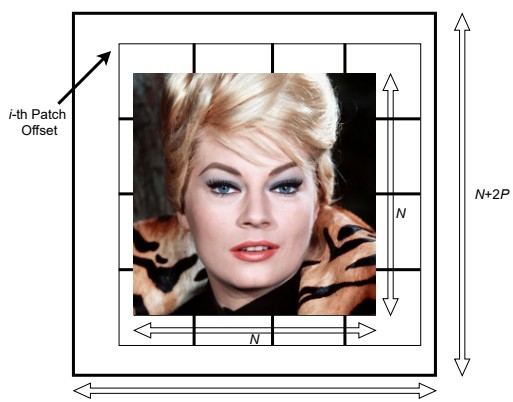

Figure 1: Schematic for zero padding and partitioning image into patches. Each index $i$ represents one of $P^2$ possible ways to choose a patch offset tuple.

For training, we use a neural network $D_{\boldsymbol{\theta}}(\boldsymbol{x}, \sigma_t)$ that accepts a noisy image $\boldsymbol{x}$ and the noise level $\sigma_t$. For each patch, we define the $x$ positional array as the 2D array consisting of the $x$ positions of each pixel of the image scaled between -1 and 1, and the $y$ positional array is similarly defined for the $y$ positions. To allow the network to learn different patch distributions at different locations in the image, we extract the corresponding patches of these positional arrays and concatenate them along the channel dimension of the noisy image patch and treat the entire array as the network input.

Since we are using a patch-based prior, we perform denoising score matching on patches of an image instead of the whole image. Hence, the training loss is given by

$$\arg\min_{\boldsymbol{\theta}} \mathbb{E}_{t\sim\mathcal{U}(0,T)}\mathbb{E}_{\boldsymbol{x}\sim p(\boldsymbol{x})}\mathbb{E}_{\boldsymbol{\varepsilon}\sim\mathcal{N}(0,\sigma_t^2 I)}\|D_{\boldsymbol{\theta}}(\boldsymbol{x}+\boldsymbol{\varepsilon},\sigma_t)-\boldsymbol{x}\|_2^2, \tag{9}$$

where $\boldsymbol{x} \sim p(\boldsymbol{x})$ represents a patch drawn from a sample of the training dataset, $\sigma_t$ is a predetermined noise schedule, and $\mathcal{U}$ denotes the uniform distribution.

## 3.2 Single measurement setting

Consider the first case where only the measurement $\boldsymbol{y}$ is given, and no in-sample training data is available. For each specific measurement $\boldsymbol{y}$, the DIP framework optimizes the network parameters $\boldsymbol{\theta}$ via the self-supervised loss (5) from the predicted reconstructed image. Diffusion models provide a prediction of the reconstructed image at each timestep: namely, the expectation of the clean image $\mathbb{E}[\boldsymbol{x}_0|\boldsymbol{x}_t]$ is approximated by the denoiser $D_{\boldsymbol{\theta}}(\boldsymbol{x}_t)$ via Tweedie's formula. Then the expectation conditioned on the measurement $\mathbb{E}[\boldsymbol{x}_0|\boldsymbol{x}_t,\boldsymbol{y}]$ can be obtained through one of many methods of enforcing the data fidelity constraint.

We begin with the unconditional expectation by leveraging the patch-based prior. Following (8), we apply Tweedie's formula to express the denoiser of $\boldsymbol{x}$ in terms of solely the denoisers of the patches of $\boldsymbol{x}$. Because the outermost product is computationally very expensive, in practice we approximate $D_{\boldsymbol{\theta}}(\boldsymbol{x})$ using only a single randomly selected value of $i$ for each denoiser evaluation:

$$D_{\boldsymbol{\theta}}(\boldsymbol{x}) \approx D_{i,B}(\boldsymbol{x}_{i,B}) + \sum_{r=1}^{(k+1)^2} D_{i,r}(\boldsymbol{x}_{i,r}). \tag{10}$$

By definition, $D_{i,B}(\boldsymbol{x}_{i,B}) = 0$ and we compute each $D_{i,r}(\boldsymbol{x}_{i,r})$ with the network. Note that (10) provides an *unconditional* estimate of the clean image; to obtain a conditional estimate $D_{\boldsymbol{\theta}}(\boldsymbol{x}_t|\boldsymbol{y})$ of the clean image, we run $C$ iterations of the conjugate gradient descent algorithm for minimizing $\|\boldsymbol{A}\boldsymbol{x} - \boldsymbol{y}\|_2$, initialized with the unconditional estimate (Chung et al., 2024). The image that is being reconstructed might not come from the distribution of the training images. Hence, the estimate $D_{\boldsymbol{\theta}}(\boldsymbol{x}_t|\boldsymbol{y})$ may be far from the true denoised image. Thus, we use $\boldsymbol{y}$ to update the parameters of the network in a way such that $D_{\boldsymbol{\theta}}(\boldsymbol{x}_t|\boldsymbol{y})$ becomes more consistent with the measurement:

$$\boldsymbol{\theta} \leftarrow \arg\min_{\boldsymbol{\theta}} \|\boldsymbol{y} - \boldsymbol{A}\,D_{\boldsymbol{\theta}}(\boldsymbol{x}_t|\boldsymbol{y})\|_2^2. \tag{11}$$

Previously, additional LoRA parameters (Hu et al., 2021) have been used as an injection to the network to leave the original parameters unchanged during this process (Barbano et al., 2023; Chung & Ye, 2024). However, the effect of using different ranks for LoRA versus other methods of network fine-tuning on DIS has not been studied extensively, so we opt to update all the weights of the network in this step. Appendix A.3 shows results from using the LoRA module.

Crucially, iterative usage of CG for computing the conditional denoiser allows for simple and efficient backpropagation through this loss function, a task that would be much more computationally challenging if another DIS such as Chung et al. (2023a) or Wang et al. (2022) were used. Furthermore, because the number of diffusion steps is large and the change in $\boldsymbol{x}_t$ is small between consecutive timesteps, we apply this network refining step only for certain iterations of the diffusion process, reducing the computational burden.

---

**Algorithm 1** Single Measurement Inverse Solver

**Require:** $\sigma_1 < \sigma_2 < \ldots < \sigma_T$, $\epsilon > 0$, $P, C, \boldsymbol{y}, K$
  Initialize $\boldsymbol{x} \sim \mathcal{N}(0, \sigma_T^2 \boldsymbol{I})$
  **for** $t = T : 1$ **do**
    **if** $t \bmod K = 0$ **then**
      Compute $D_{\boldsymbol{\theta}}(\boldsymbol{x}_t)$ using (10) with
        a random index $i$
      Run $C$ iterations of CG initialized
        with $D_{\boldsymbol{\theta}}(\boldsymbol{x}_t)$ to obtain $D_{\boldsymbol{\theta}}(\boldsymbol{x}_t|\boldsymbol{y})$
      Define $L(\boldsymbol{\theta}) = \|\boldsymbol{y} - \boldsymbol{A}D_{\boldsymbol{\theta}}(\boldsymbol{x}_t|\boldsymbol{y})\|_2^2$
      Update $\boldsymbol{\theta}$ by backpropagating $L(\boldsymbol{\theta})$
    **end if**
    Sample $\boldsymbol{z} \sim \mathcal{N}(0, \sigma_t^2 \boldsymbol{I})$
    Set $\alpha_t = \epsilon \cdot \sigma_t^2$
    Compute $D(\boldsymbol{x}_t)$ using (10) with a
      random index $i$
    Run $C$ iterations of CG for (7)
      initialized with $D(\boldsymbol{x}_t)$
    Set $\boldsymbol{s}_t = (D - \boldsymbol{x}_t)/\sigma_t^2$
    Set $\boldsymbol{x}_{t-1}$ to $\boldsymbol{x}_t + \frac{\alpha_t}{2}\boldsymbol{s}_t + \sqrt{\alpha_t}\boldsymbol{z}$
  **end for**

---

After this step, we apply the refined network to compute a new estimate of the score of $\boldsymbol{x}_t$ and then use it to update $\boldsymbol{x}_t$. Similar to the network refining step, we use the stochastic version of the denoiser given by (10) rather than the full version. Hu et al. (2024) showed that for patch-based priors, Langevin dynamics Song & Ermon (2019) works particularly well as a sampling algorithm, so we use it here in conjunction with CG steps to enforce data fidelity. Algorithm 1 summarizes the entire method for cases where only a single measurement $\boldsymbol{y}$ is available.

### 3.3 SMALL DATASET SETTING

Now turn to the case where we have trained a diffusion model on OOD data, but we also have a very small dataset of in-distribution test data that we can use to fine-tune the model. When fine-tuning, we initialize the network with the checkpoint trained on OOD data and then use the denoising score matching loss function to fine-tune the network on in-distribution data. Wang et al. (2023) found that improved image generation performance can be obtained by training with varying patch sizes, as opposed to fixing the patch size to the one used during inference. Here, we apply a varying patch size scheme during fine-tuning also as a method of data augmentation. We use the UNet architecture in Ho et al. (2020) that can accept images of different sizes. Hence, the loss becomes

$$\arg \min_{\boldsymbol{\theta}} \mathbb{E}_{t \sim \mathcal{U}(0,T)} \mathbb{E}_{\boldsymbol{x} \sim p_d(\boldsymbol{x})} \mathbb{E}_{\boldsymbol{\varepsilon} \sim \mathcal{N}(0,\sigma_t^2 I)} \|D_{\boldsymbol{\theta}}(\boldsymbol{x} + \boldsymbol{\varepsilon}, \sigma_t) - \boldsymbol{x}\|_2^2, \tag{12}$$

where $\boldsymbol{x} \sim p_d(\boldsymbol{x})$ represents the drawing a randomly sized patch from an image belonging to the fine-tuning dataset. Appendix A.5 provides full details of the training process.

At reconstruction time, we assume that our network has been fine-tuned reasonably to our dataset. Thus, we remove the network refining step in Algorithm 1 and keep the weights fixed throughout the entire process. We still use the same CG descent algorithm to enforce data fidelity with the measurement.

## 4 EXPERIMENTS

**Experimental setup.** For the CT experiments, we used the AAPM 2016 CT challenge data from McCollough et al. (2017). We applied the same data processing methods as in Hu et al. (2024) with the exception that we used all the XY slices from the 9 training volumes to train the in distribution networks, yielding a total of approximately 5000 slices. For the deblurring and superresolution experiments, we used the CelebA-HQ dataset (Liu et al., 2015) with each image having size $256 \times 256$. The test data was a randomly selected subset of 10 of the images not used for training. In all cases, we report the average metrics across the test images: peak SNR (PSNR) in dB, and structural similarity metric (SSIM) (Wang et al., 2004). For the training data, we trained networks on generated phantom images consisting of randomly placed ellipses of different shapes and sizes. See Fig. 20 for examples. These phantoms can be generated on the fly in large quantities. We used networks trained on grayscale phantoms for the CT experiments and networks trained on RGB phantoms for the deblurring and superresolution experiments. Appendix A.4 contains precise specifications of the phantoms.

We trained the patch-based networks with $64 \times 64$ patches and used a zero padding value of 64, so that 5 patches in both directions were used to cover the target image. We used the network architecture in Karras et al. (2022) for both the patch-based networks and whole-image networks. All networks were trained on PyTorch using the Adam optimizer with 2 A40 GPUs.

**Single measurement setting.** In cases where no training data is available and we only have the measurement $\boldsymbol{y}$, we applied Algorithm 1 to solve a variety of inverse problems: CT reconstruction, deblurring, and superresolution. For the forward and backward projectors in CT reconstruction, we used the implementation provided by the ODL Team (2022). We performed two sparse-view CT (SVCT) experiments: one using 20 projection views, and one using 60 projection views. Both of these were done using a parallel beam forward projector where the detector size was 512 pixels. For the deblurring experiments, we used a uniform blur kernel of size $9 \times 9$ and added white Gaussian noise with $\sigma = 0.01$ where the clean image was scaled between 0 and 1. For the superresolution ex-

periments, we used a scaling factor of 4 with downsampling by averaging and added white Gaussian noise with $\sigma = 0.01$.

For the comparison methods, we ran experiments that naively used the OOD diffusion model without the self-supervised network refining process. For reference, we also ran experiments using a diffusion model trained on the entire in-distribution training set (the "correct" model). In practice, it would not be possible to obtain such a large training dataset of in-distribution images. Additionally, for these diffusion model methods, we implemented both the patch-based version as well as the whole-image version. The whole-image networks were trained with the loss function in (9) and used the same network architecture as the patch-based models, but the input of the network was the entire image and did not contain positional encoding information.

We also compared with more traditional methods: applying a simple baseline, reconstructing via the total variation regularizer (ADMM-TV), and two plug and play (PnP) methods: PnP-ADMM (Xu et al., 2020) and PnP-RED (Hu et al., 2022). For CT, the baseline was obtained by applying the filtered back-projection method to the measurement $y$. For deblurring, the baseline was simply equal to the blurred image. For superresolution, the baseline was obtained by upsampling the low resolution image and using nearest neighbor interpolation. The implementation of ADMM-TV can be found in Hong et al. (2024a). Finally, since we assume we do not have access to any clean training data, we used the off the shelf denoiser BM3D (Dabov et al., 2006). Appendix A.5 contains the values of all the parameters of the algorithms.

Table 1 shows the main results for single-measurement inverse problem solving. The bottom two rows show the hypothetical performance if it were possible to train a diffusion model on a large dataset of in distribution images, which is not available in practice. Our self-supervised patch-based diffusion approach achieved significantly higher quantitative results when averaged across the test dataset than the self-supervised whole-image approach in all the inverse problems. Furthermore, although the diffusion model that was initially used in this algorithm was trained on completely different images, by applying the self-supervised loss, the patch-based approach is able to achieve results that are close to (and for the deblurring case, even surpassing) those using the in-distribution networks. The table also shows that by including the self-supervised step, a dramatic improvement over naively using the OOD model is achieved. Lastly, Fig. 2 shows that some artifacts appear in the whole-image SS method that are not present in our patch SS method.

Table 1: Comparison of quantitative results on three different inverse problems in the single measurement setting. Results are averages across all images in the test dataset. Best results for practical use are in bold.

| Method | CT, 20 Views | | CT, 60 Views | | Deblurring | | Superresolution | |
|---|---|---|---|---|---|---|---|---|
| | PSNR↑ | SSIM↑ | PSNR↑ | SSIM↑ | PSNR↑ | SSIM↑ | PSNR↑ | SSIM↑ |
| Baseline | 24.93 | 0.613 | 30.15 | 0.784 | 23.93 | 0.666 | 25.42 | 0.724 |
| ADMM-TV | 26.81 | 0.750 | 31.14 | 0.862 | 27.58 | 0.773 | 25.22 | 0.729 |
| PnP-ADMM (Xu et al., 2020) | 30.20 | 0.838 | 36.75 | 0.932 | 28.98 | 0.815 | 27.29 | 0.796 |
| PnP-RED (Hu et al., 2022) | 27.12 | 0.682 | 32.68 | 0.876 | 28.37 | 0.793 | 27.73 | 0.809 |
| Whole image, naive | 28.11 | 0.800 | 33.10 | 0.911 | 25.85 | 0.742 | 25.65 | 0.742 |
| Patches, naive (Hu et al., 2024) | 27.44 | 0.719 | 33.97 | 0.934 | 26.77 | 0.782 | 26.12 | 0.759 |
| Self-supervised, whole (Barbano et al., 2023) | 33.19 | 0.861 | 40.47 | 0.957 | 29.50 | 0.831 | 27.07 | 0.701 |
| Self-supervised, patch (Ours) | **33.77** | **0.874** | **41.45** | **0.969** | **30.34** | **0.860** | **28.10** | **0.827** |
| Whole image, correct* | 33.99 | 0.886 | 41.67 | 0.969 | 29.87 | 0.851 | 28.33 | 0.801 |
| Patches, correct* | 34.02 | 0.889 | 41.70 | 0.967 | 30.12 | 0.865 | 28.49 | 0.835 |

*not available in practice for mismatched distribution inverse problems

To demonstrate that our method also works well even when the mismatched distribution is closer to the true distribution, we also ran an experiment where the networks were initially trained on the LIDC-IDRI dataset (Armato et al., 2011). We extracted 10000 2D slices from the 3D volumes and rescaled all the images so that the pixel values were between 0 and 1. We then ran Algorithm 1 to perform CT reconstruction where the test dataset was the same as the one used in Table 1. Table 4 shows the results of this experiment. Our method achieved better quantitative results than the whole image method and even outperformed the reconstructions using the in distribution network but without any self-supervision. Appendix A.1 shows the visual results of these experiments. Appendix A.2 further discusses using self-supervision in cases where the initial network was trained on in-distribution data and shows improved image quality.

We also ran ablation studies to examine the effect of various parameters on the proposed method. Barbano et al. (2023) and Chung & Ye (2024) used the LoRA module for solving single-measurement inverse problems with diffusion models. We tested this method for CT reconstruction and deblurring with different rank adjustments and found this method to be inferior to modifying the weights of the entire network. We also ran experiments using networks with different numbers of weights. Appendix A.3 shows the results of these experiments.

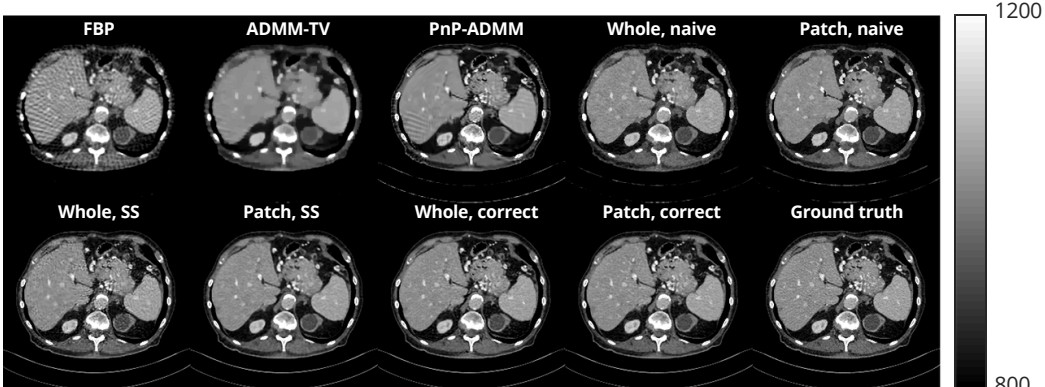

Figure 2: **Single measurement setting**: Results of 60 view CT reconstruction using self supervised (SS) approach. The display uses modified HU units to show more contrast between organs.

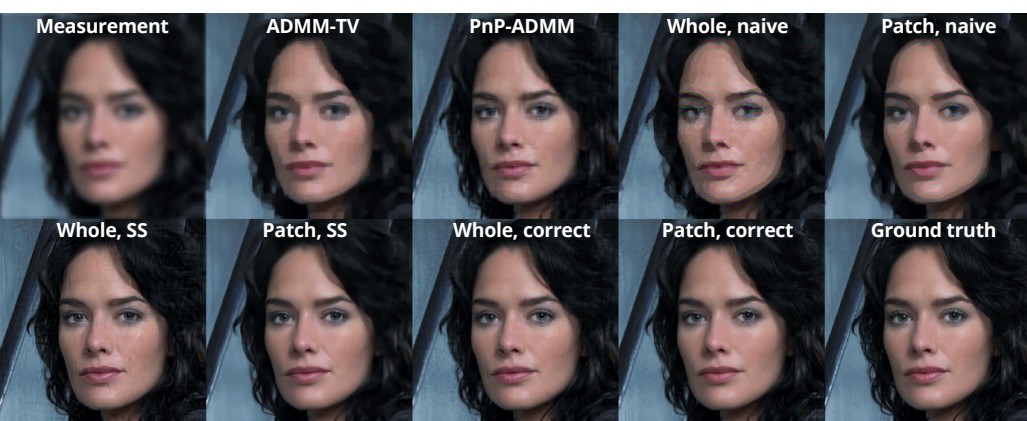

Figure 3: **Single measurement setting**: Results of deblurring using self supervised (SS) approach and comparison methods.

**Small dataset setting.** We ran experiments on the same inverse problems as the single measurement case. The OOD networks were fine-tuned with 10 images randomly selected from the in-distribution training set; we also ran ablation studies using different quantities of in-distribution data in Appendix A.3. Figures 4 and 5 show that the patch-based model is much less prone to overfitting than the whole-image model. Hence, to evaluate the best possible performance of the whole-image model compared to the patch-based model, for both models we chose the checkpoint yielding the best results for solving inverse problems.

Table 2 shows the main results for solving inverse problems using the fine-tuned diffusion model. We compared the results of fine-tuning the whole-image model with fine-tuning the patch-based model as well as the best baseline out of the four baselines shown in Table 1, namely "baseline", ADMM-TV, PnP-ADMM, and PnP-RED. The results show that the proposed patch-based method achieved the best performance in terms of quantitative metrics for all of the inverse problems. Figure 6 shows the visual results of this experiment. The patch-based model is able to learn an acceptable prior using the very small in-distribution dataset and the reconstructed images contain fewer artifacts than the comparison methods.

Table 2: Comparison of results for using diffusion models fine-tuned on 10 in-distribution images to solve inverse problems in small dataset setting. Best results are in bold.

| Method | CT, 20 Views | | CT, 60 Views | | Deblurring | | Superresolution | |
|---|---|---|---|---|---|---|---|---|
| | PSNR↑ | SSIM↑ | PSNR↑ | SSIM↑ | PSNR↑ | SSIM↑ | PSNR↑ | SSIM↑ |
| Best baseline | 30.20 | 0.838 | 36.75 | 0.932 | 28.98 | 0.815 | 27.73 | 0.809 |
| Whole image | 33.09 | **0.875** | 40.54 | 0.964 | 28.41 | 0.812 | 27.29 | 0.775 |
| Patches (Ours) | **33.44** | **0.875** | **41.21** | **0.965** | **29.25** | **0.840** | **28.10** | **0.827** |
| Patches, correct* | 34.02 | 0.889 | 41.70 | 0.967 | 30.12 | 0.865 | 28.49 | 0.835 |

*not available in practice for mismatched distribution inverse problems

Figures 4 and 5 further investigates the effect of overfitting. For different amounts of training time using the small in-distribution dataset, we ran the reconstruction algorithm for 60-view CT. While the whole-image model exhibited substantial image degradation when the network was fine-tuned for too long, the patch-based model retained relatively stable performance throughout the entire training process. This illustrates that whole-image diffusion models exhibits severe overfitting problems when only a small amount of training data is unavailable. Furthermore, patch-based diffusion models assist greatly with this problem and the results are evident for solving inverse problems. Appendix A.1 shows the visual results of these experiments.

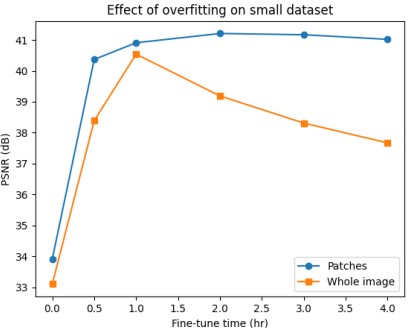

Figure 4: Comparison of PSNR between patch-based model and whole-image model for overfitting in small dataset setting.

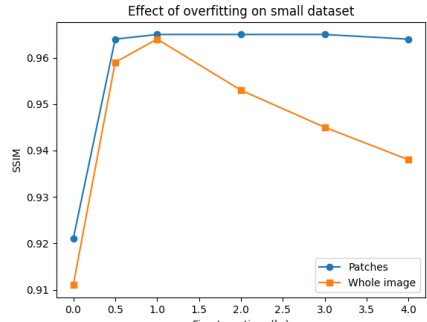

Figure 5: Comparison of SSIM between patch-based model and whole-image model for overfitting in small dataset setting.

To look at the priors learned by the different models from fine-tuning, we unconditionally generated images from the checkpoints obtained by fine-tuning on the 10 image CT dataset. Figure 7 shows a subset of the generated images where we used the checkpoints obtained after 4 hours of training. The top two rows consist of images generated by the whole-image model and the bottom two rows consist of images generated by the patch diffusion model. To emphasize the memorization point, we grouped together similar looking images in the top two rows: it can be seen that the images in each group look virtually identical, despite the fact that the pure white noise initializations for each sample was different. On the other hand, while the samples generated by the patch diffusion model also show some unrealistic features, they all show some distinct features, which implies that this model has much better generalization ability.

## 5 CONCLUSION

This paper presented a method of using patch-based diffusion models to solve inverse problems when the data distribution is mismatched from the trained network. In particular, we conducted experiments in the setting when only a single measurement is available as well as the setting when a very small subset of in-distribution data is available. In both settings, the proposed patch-based method outperformed whole-image methods in a variety of inverse problems. In the future, more work could be done on using acceleration methods for faster reconstruction, exploring other less computationally expensive methods of fine-tuning the network geared toward inverse problem solving, and methods of refining the prior when a set of measurements are available (Yaman et al., 2020).

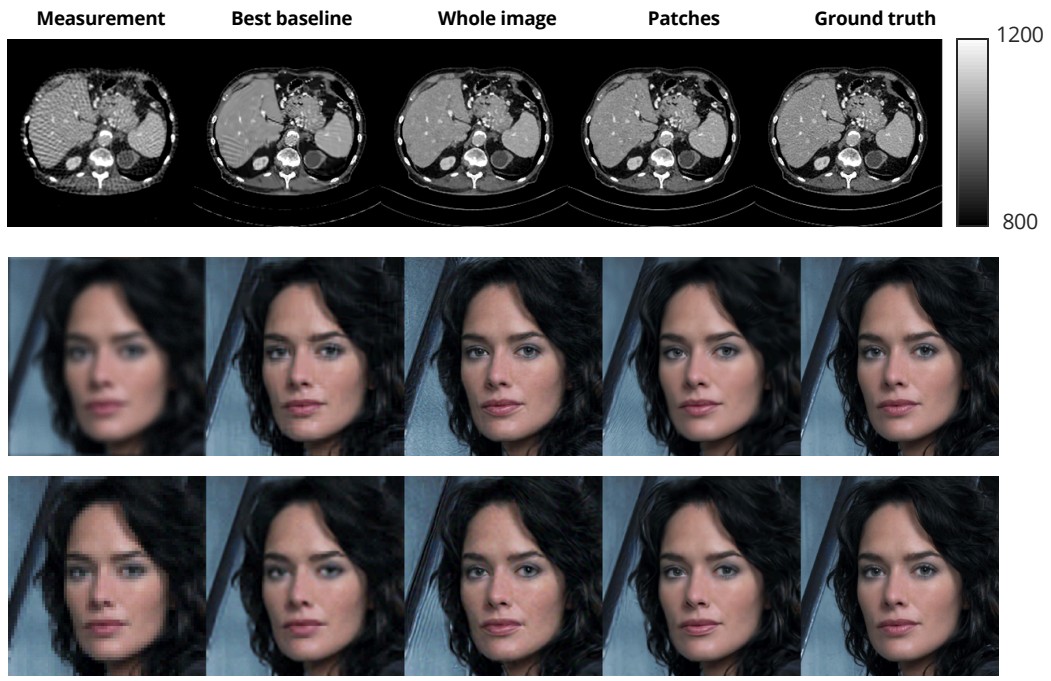

Figure 6: **Small dataset setting**: Results of inverse problem solving. Top row is 60 view CT recon, middle row is deblurring, and bottom row is superresolution. For CT, measurement refers to FBP.

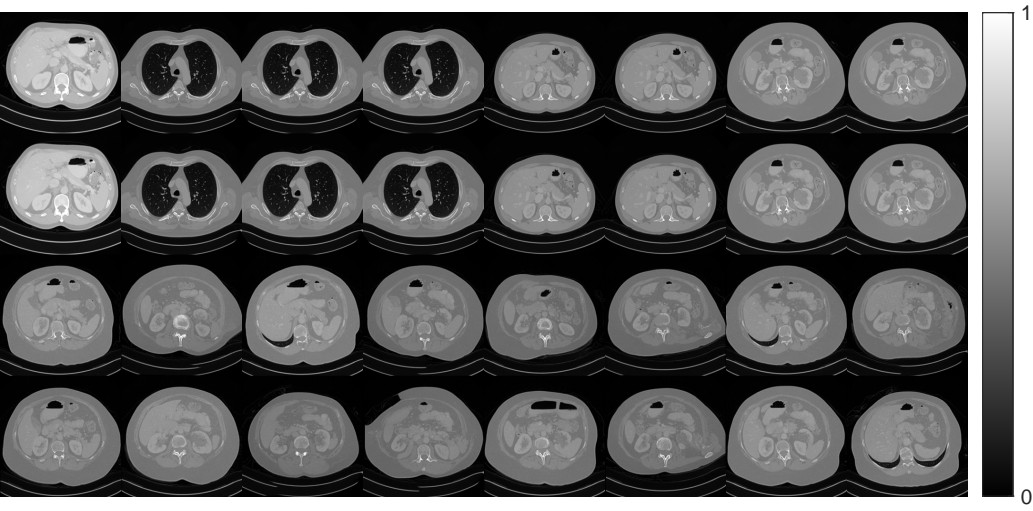

Figure 7: Unconditional generation of CT images from networks fine-tuned in the small dataset setting. Top two rows were generated with the whole image model; bottom two rows were generated with the patch-based model.

Limitations of the work include a slow runtime for the self-supervised algorithm and a lack of theoretical guarantees for dataset size requirements. Providing uncertainty quantification is also an open problem for such self-supervised methods.

ACKNOWLEDGMENTS

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

# A APPENDIX

## A.1 ADDITIONAL INVERSE PROBLEM SOLVING FIGURES

Figure 8 shows the results of various methods applied to superresolution in the single measurement setting.

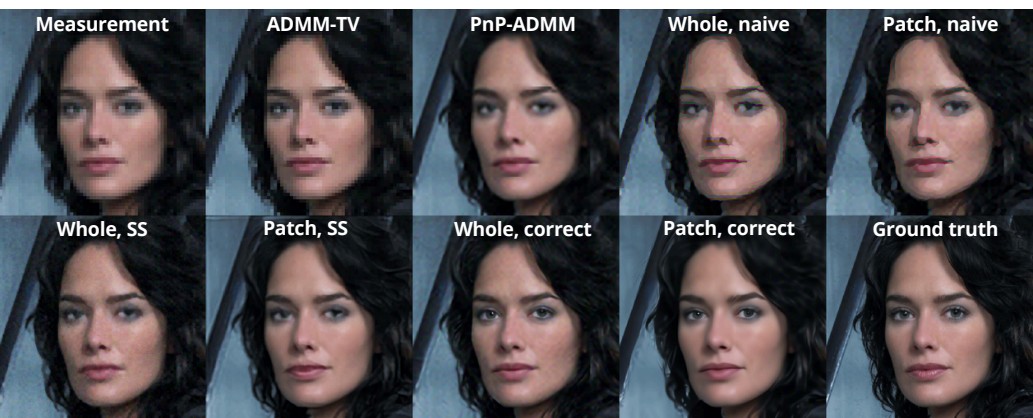

Figure 8: Results of superresolution using self supervised (SS) approach and comparison methods.

Figure 9 shows the results of 20 view CT reconstruction using Algorithm 1. This very sparse view CT recon problem is made more challenging by the lack of any training data. Artifacts can clearly be seen in all the comparison methods. Despite this challenge, reconstructions such as this one can still be useful for medical applications such as patient positioning.

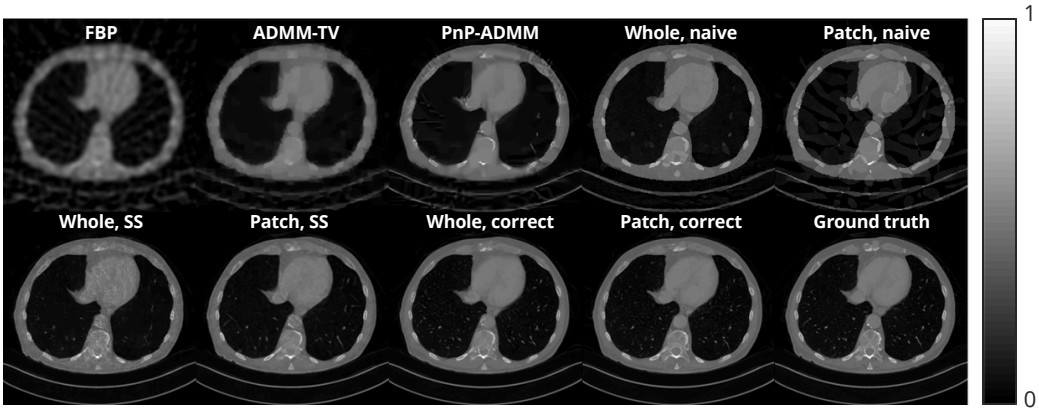

Figure 9: Results of 20 view CT reconstruction in a self-supervised setting. For clarity, the images are plotted on the same scale as the diffusion models were trained.

Figure 10 shows the results of running self-supervised CT reconstruction with 20 views and 60 views where the starting checkpoint was obtained through training on a large (but out of distribution) CT dataset: 10000 LIDC-IDRI slices. Particularly for 20 views, the artifacts from using the whole image model are apparent, while the patch-based model obtains a much higher quality reconstruction. Thus, regardless of whether the starting network has a severely mismatched distribution (ellipses) or a slightly mismatched distribution (different CT dataset), our proposed method outperforms the whole image model.

Figure 11 shows the results of performing 60 view CT reconstruction in an unsupervised manner from checkpoints fine-tuned using the small in distribution CT dataset. The images on the bottom row shows the progressively worsening degradation and increasing number of artifacts resulting

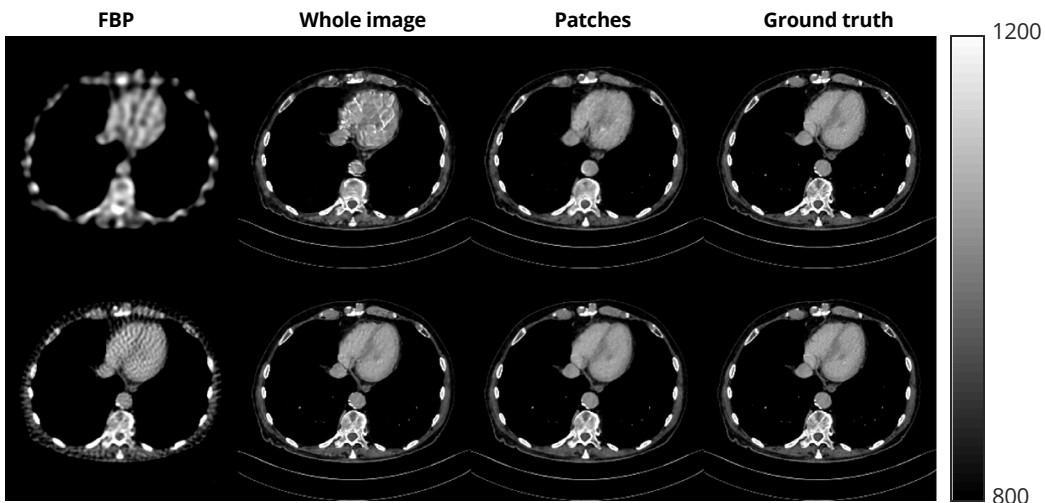

Figure 10: Results of CT reconstruction in a self-supervised setting when the starting network was trained on the LIDC dataset. Top row used 20 views and bottom row used 60 views.

from overfitting exhibited by whole image model. On the other hand, the top row shows relatively stable performance exhibited by the patch-based model as it is able to avoid overfitting much better.

Table 3: Performance of fine-tuning on 60 view CT using checkpoints trained for different lengths of time. Best results are in bold.

| Train | Patches | | Whole image | |
|---|---|---|---|---|
| time (hr) | PSNR↑ | SSIM ↑ | PSNR↑ | SSIM ↑ |
| 0 | 33.91 | 0.921 | 33.10 | 0.911 |
| 0.5 | 40.37 | 0.964 | 38.39 | 0.959 |
| 1 | 40.91 | **0.965** | **40.54** | **0.964** |
| 2 | **41.21** | **0.965** | 39.19 | 0.953 |
| 3 | 41.17 | **0.965** | 38.31 | 0.945 |
| 4 | 41.02 | 0.964 | 37.67 | 0.938 |

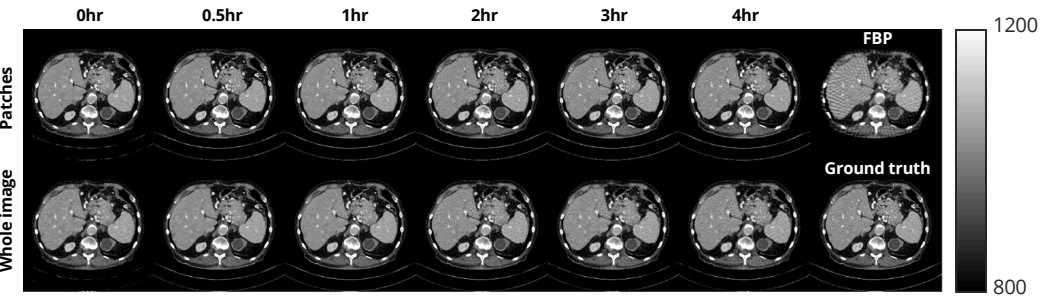

Figure 11: Results of 60 view CT recon with networks fine-tuned on 10 in distribution CT images for varying amounts of training time.

## A.2 EFFECT OF SELF-SUPERVISION FOR DIFFERENT DISTRIBUTIONS

Recall that in the single measurement setting, Algorithm 1 is used to adjust the underlying distribution of the network away from the originally trained OOD data and toward the ground truth image. We investigated the effect of applying this method even when the network was trained on the

in-distribution data. Figures 12 and 13 show the results of this experiment for CT reconstruction, where each point represents the specific PSNR for one of the images in the test dataset. If the additional self-supervision step had no effect on the image quality, the points would lie on the red line. However, in both cases, all of the points are above the red line, indicating that the self-supervision step of the algorithm improves the image quality even when the network was already trained on in-distribution data. Furthermore, the improvement is more substantial for for the 20 view case than the 60 view case, as the predicted clean images $D_\theta(x_t|y)$ at each step for the 60 view case are likely to be more closely aligned with the measurement, so the network refining step becomes less significant. Importantly, this shows that in practice, one may directly apply Algorithm 1 to solve inverse problems without knowledge of the severity of the mismatch in distribution between training and testing data: even when there is no mismatch, the additional self-supervision step does not degrade the image quality.

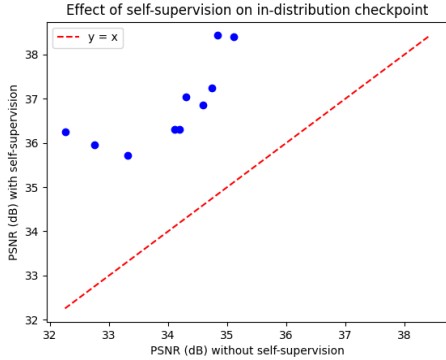
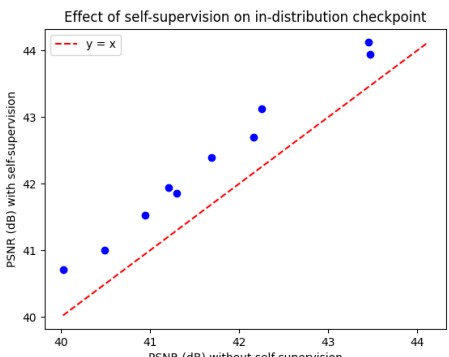

Figure 12: PSNR of 20 view CT reconstruction in single-measurement setting using a patch-based in-distribution network.

Figure 13: PSNR of 60 view CT reconstruction in single-measurement setting using a patch-based in-distribution network.

Table 5 summarizes the results of using different training datasets while keeping the same test dataset (AAPM CT images). The distribution shift is greatest when the network is trained on ellipse phantoms and used to reconstruct the AAPM CT images, so the reconstruction quality is the lowest in this case. The LIDC dataset consists of CT images which belong to a distribution that is reasonably similar to the distribution of AAPM CT images, so when using the network trained on LIDC images, the quality drop over using an in-distribution network is not substantial. Finally, the improvements obtained by using more in-distribution networks is more apparent for the 20 view case as the measurements are sparser for this case, so the prior plays a larger role in obtaining an accurate reconstruction.

Table 4: Single measurement CT reconstruction results where the initial checkpoint was trained on LIDC dataset and refined on the fly with the AAPM measurement.

| Dataset | CT, 20 views | | CT, 60 views | |
| size | PSNR↑ | SSIM ↑ | PSNR↑ | SSIM ↑ |
|---|---|---|---|---|
| Whole image | 35.01 | 0.894 | 41.95 | 0.967 |
| Patches (Ours) | 36.34 | 0.918 | 42.32 | 0.972 |

Table 5: Performance of patch-based model in single measurement setting for CT reconstruction for different OOD training datasets.

| Train time (hr) | 20 views | | 60 views | |
|---|---|---|---|---|
| | PSNR↑ | SSIM ↑ | PSNR↑ | SSIM ↑ |
| Ellipses | 33.77 | 0.874 | 41.45 | 0.966 |
| LIDC | 36.34 | 0.918 | 42.32 | 0.970 |
| AAPM | 36.82 | 0.923 | 42.33 | 0.970 |

## A.3 ABLATION STUDIES

We performed four ablation studies to evaluate the impact of various parameters on the proposed methods. Similar to the main text, all quantitative results are averaged across the test dataset.

**Low rank adaptation.** To avoid overfitting to the measurement in self-supervised settings, Barbano et al. (2023) proposed using a low rank adaptation to the weights of the neural network, reducing the number of weights that are adjusted during reconstruction by a factor of around 100. Here we investigate the effect of using different ranks of adaptations on two inverse problems: 60 view CT reconstruction and deblurring. Consistent with Barbano et al. (2023) and Chung & Ye (2024), we only used the LoRA module for attention and convolution layers. We also allowed the biases of the network to be changed.

Tables 6 and 7 show the quantitative results of these experiments, where a rank of "full" represents fine-tuning all the weights of the network. In all cases, using LoRA for this fine-tuning process results in worse reconstructions than simply fine-tuning the entire network. The visual results are especially apparent in Figure 15: the reconstructed image becomes oversmoothed when using LoRA and artifacts become present when using the whole image model. This is likely due to the large distribution shift between the initial distribution of images and target distribution of faces: the low rank adaptation to the mismatched network is not sufficient to represent the new distribution and thus the self-supervised loss function results in smoothed images.

Table 6: Performance of 60 view CT recon using self-supervised network refining with LoRA module. Best results are in bold.

| Rank | Parameters (%) | Patches | | Whole image | |
|---|---|---|---|---|---|
| | | PSNR↑ | SSIM ↑ | PSNR↑ | SSIM ↑ |
| 2 | 1.1 | 40.37 | 0.963 | 39.25 | 0.952 |
| 4 | 2.0 | 40.32 | 0.963 | 39.10 | 0.951 |
| 8 | 3.8 | 40.33 | 0.963 | 39.18 | 0.951 |
| 16 | 7.2 | 40.32 | 0.963 | 39.33 | 0.953 |
| Full | 100 | **41.45** | **0.966** | **40.47** | **0.957** |

Table 7: Performance of deblurring using self-supervised network refining with LoRA module. Best results are in bold.

| Rank | Parameters (%) | Patches | | Whole image | |
|---|---|---|---|---|---|
| | | PSNR↑ | SSIM ↑ | PSNR↑ | SSIM ↑ |
| 2 | 1.1 | 29.31 | 0.830 | 29.19 | 0.811 |
| 4 | 2.0 | 29.31 | 0.829 | 29.35 | 0.817 |
| 8 | 3.8 | 29.38 | 0.831 | 29.19 | 0.810 |
| 16 | 7.2 | 29.31 | 0.830 | 29.33 | 0.815 |
| Full | 100 | **30.34** | **0.860** | **29.50** | **0.831** |

**Effect of network size.** In the self-supervised case, another potential method to avoid overfitting is to use a smaller network. We trained networks with differing numbers of base channels (but no other

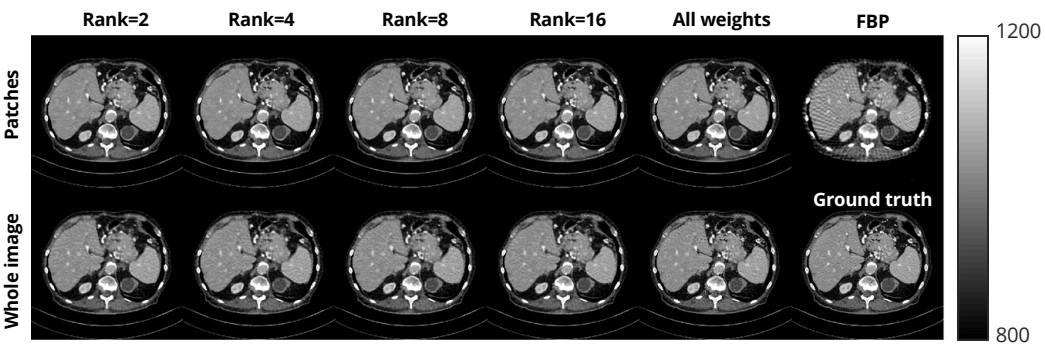

Figure 14: Results of using LoRA module for 60 view CT reconstruction in a single measurement setting. All weights refers to adjusting all the weights of the network at reconstruction time.

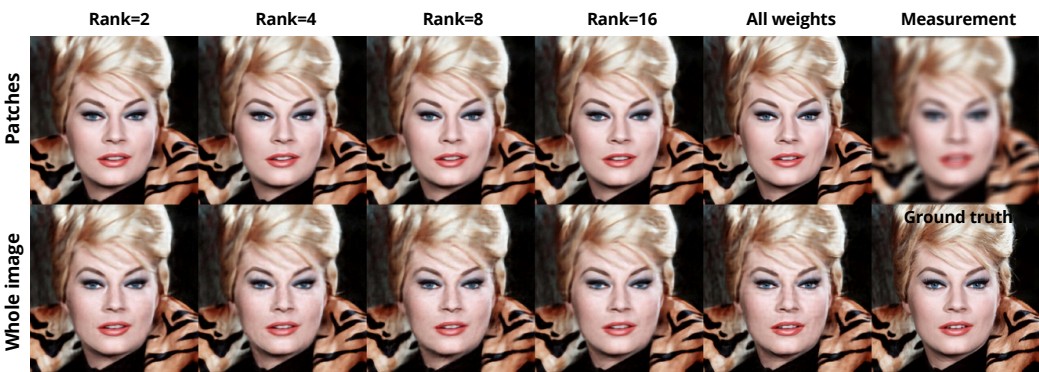

Figure 15: Results of using LoRA module for deblurring in a single measurement setting. All weights refers to adjusting all the weights of the network at reconstruction time.

modifications) on the ellipse phantom dataset and then used Algorithm 1 to perform self-supervised 60 view CT reconstruction. Table 8 shows the quantitative results of this experiment. For both the patch-based model and the whole image model, the network with 128 base channels obtained the best result, so we used this network architecture for all the main experiments. Figure 16 again shows evidence of overfitting in the form of artifacts in the otherwise smooth regions of the organs when using the network with 256 base channels. These artifacts are less obvious in the patch-based model.

Table 8: Performance of 60 view CT recon in a self-supervised manner with networks of different sizes. Best results are in bold.

| Base Channels | Parameters (Millions) | Patches PSNR↑ | SSIM ↑ | Whole image PSNR↑ | SSIM ↑ |
|---|---|---|---|---|---|
| 32 | 3.4 | 39.73 | 0.958 | 39.69 | 0.957 |
| 64 | 14 | 40.37 | 0.961 | 40.07 | **0.958** |
| 128 | 60 | **41.45** | **0.966** | **40.47** | 0.957 |
| 256 | 217 | 40.29 | 0.959 | 39.28 | 0.954 |

**Fine-tuning with a larger dataset.** To examine the effect of fine-tuning the networks on differing sizes of in-distribution datasets, we started with the same checkpoint trained on ellipses and fine-tuned them using various sizes of datasets consisting of CT images. Each small dataset consisted of randomly selected images from the entire 5000 image AAPM dataset. Next we used these checkpoints to perform 60 view CT reconstruction (without any self supervision). Table 9 shows the results of these experiments, where we also included the results of using the in-distribution network

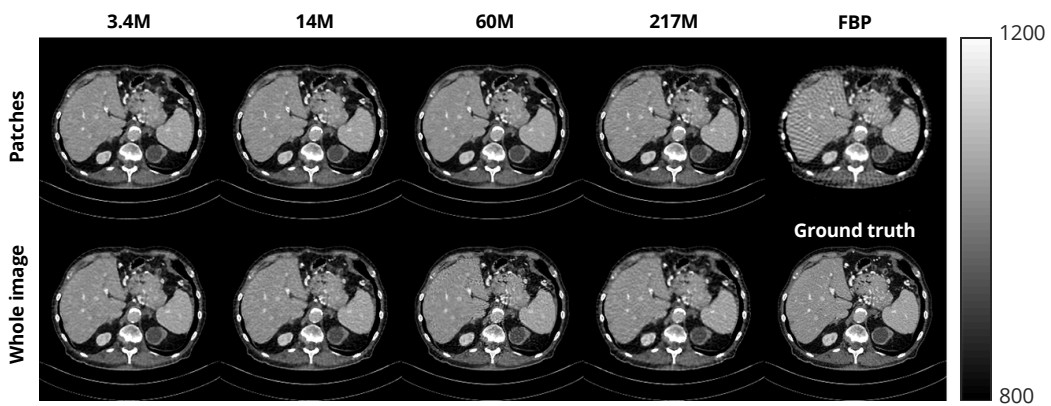

Figure 16: Results of 60 view CT recon using networks with different numbers of parameters in the single-measurement setting. The top numbers show the number of total parameters in the network.

trained on the entire 5000 image dataset. This shows that for a wide range of different fine-tuning dataset sizes our proposed method obtained better metrics than the whole-image model.

We emphasize the difference between the results of Hu et al. (2024), which showed that patch-based models outperform whole image models in cases of limited data, and the results here. Since the networks in Hu et al. (2024) were trained from scratch, more data was required: the smallest datasets used in Hu et al. (2024) contained 144 images. In constrast, we are able to fine-tune networks in our work using only 10 images. Consequently, the training time is also much lower for our work: Figure 4 shows that we fine-tuned a patch-based model in only about 2 hours, whereas Hu et al. (2024) required 12-24 hours to train the patch-based models from scratch. Thus, our results complement the work of Hu et al. (2024) by showing that, compared to whole-image models, patch-based diffusion models easier to train from scratch in settings of limited data, and they are also easier to fine-tune when data is very limited.

Table 9: Performance of fine-tuning on 60 view CT using checkpoints fine-tuned from different dataset sizes. Best results are in bold.

| Dataset size | Patches | | Whole image | |
|---|---|---|---|---|
| | PSNR↑ | SSIM ↑ | PSNR↑ | SSIM ↑ |
| 3 | 40.93 | 0.964 | 40.45 | 0.964 |
| 10 | 41.21 | 0.965 | 40.54 | 0.964 |
| 30 | 41.31 | 0.966 | 40.66 | 0.967 |
| 100 | 41.46 | 0.967 | 40.96 | 0.968 |
| 5000* | **41.70** | **0.967** | **41.67** | **0.969** |

**Backpropagation iterations during self-supervision.** In the single measurement setting, the self-supervised loss is crucial to ensuring that the OOD network output is consistent with the measurement. Backpropagation through the network is necessary to minimize this loss, but too much network refining during this step could lead to overfitting to the measurement and image degradation. We ran experiments examining the effect of the number of backpropagation iterations during each step for the patch-based model and the whole image model. Figures 18 and 19 show that in both cases, performance generally improved when increasing the number of backpropagation iterations and overfitting is avoided. Additionally, the patch-based model always outperformed the whole image model and exhibited more improvement as the number of backpropagation iterations increased. For our main experiments, we used 5 iterations as the improved performance became marginal compared to the extra runtime.

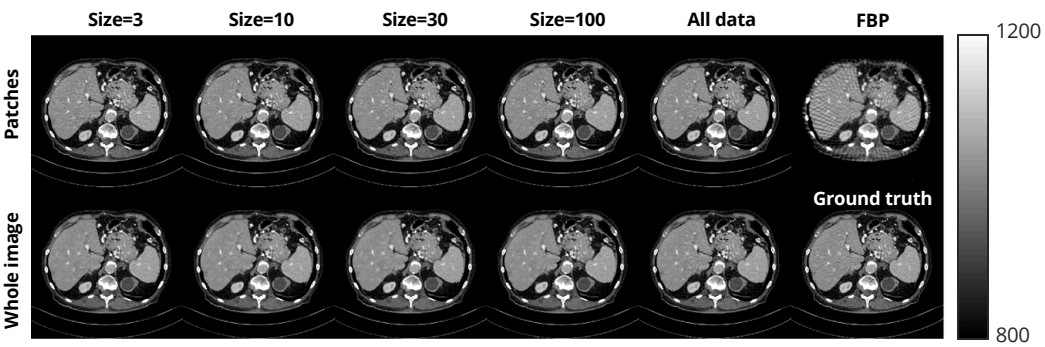

Figure 17: Results of 60 view CT recon in the small dataset setting where the size of the small dataset is varied.

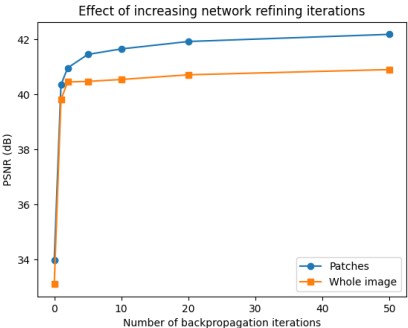

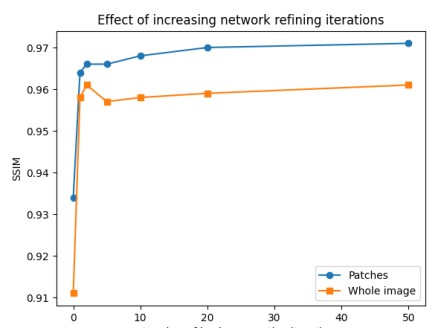

Figure 18: Comparison of PSNR between patch-based model and whole-image model for number of network refining iterations in single measurement setting.

Figure 19: Comparison of SSIM between patch-based model and whole-image model for number of network refining iterations in single measurement setting.

### A.4 PHANTOM DATASET DETAILS

We used two phantom datasets of 10000 images each: one consisting of grayscale phantoms and the other consisting of colored phantoms. The grayscale phantoms consisted of 20 ellipses with a random center within the image, each with minor and major axis having length equal to a random number chosen between 2 and 20 percent of the width of the image. The grayscale value of each ellipse was randomly chosen between 0.1 and 0.5; if two or more ellipses overlapped, the grayscale values were summed for the overlapped area with all values exceeding 1 set to 1. Finally, all ellipses were set to a random angle of rotation. The colored phantoms were generated in the same way, except the RGB values for each ellipse were set independently and then multiplied by 255 at the end. Figure 20 shows some of the sample phantoms.

### A.5 EXPERIMENT PARAMETERS

We applied the framework of Karras et al. (2022) to train the patch-based networks and whole image networks. Since images were scaled between 0 and 1 for both grayscale images and RGB channels, we chose a maximum noise level of $\sigma = 40$ and a minimum noise level of $\sigma = 0.002$ for training. We used the same UNet architecture for all the networks consisting of a base channel multiplier size of 128 and 2, 2, and 2 channels per resolution for the three layers. We also used dropout connections with a probability of 0.05 and exponential moving average for weight decay with a half life of 500K images to avoid overfitting.

The learning rate was chosen to be $2 \cdot 10^{-4}$ when training networks from scratch and was $1 \cdot 10^{-4}$ for the fine-tuning experiments. For the patch-based networks, the batch size for the main patch size ($64 \times 64$) was 128, although batch sizes of 256 and 512 were used for the two smaller patch sizes

Table 10: Performance of Algorithm 1 for 60 view CT reconstruction in single measurement setting with different numbers of backpropagation iterations. Best results are in bold.

| Backprop iterations | Patches | | Whole image | |
|---|---|---|---|---|
| | PSNR↑ | SSIM ↑ | PSNR↑ | SSIM ↑ |
| 0 | 33.97 | 0.934 | 33.10 | 0.911 |
| 1 | 40.35 | 0.964 | 39.81 | 0.958 |
| 2 | 40.96 | 0.966 | 40.45 | 0.961 |
| 5 | 41.45 | 0.966 | 40.47 | 0.957 |
| 10 | 41.65 | 0.968 | 40.54 | 0.958 |
| 20 | 41.92 | 0.970 | 40.71 | 0.959 |
| 50 | **42.18** | **0.971** | **40.90** | **0.961** |

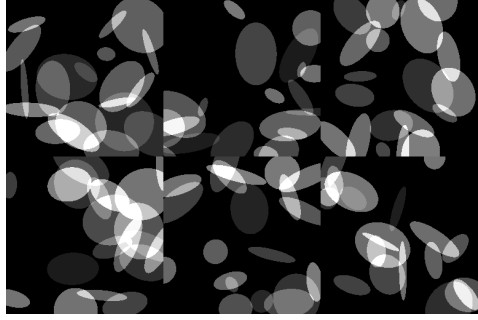
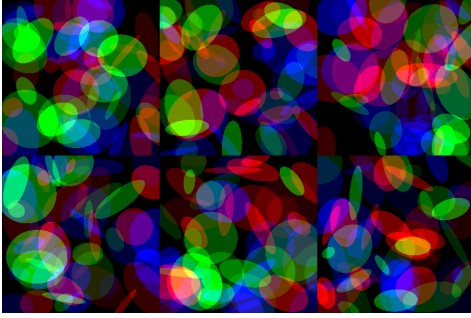

(a) Six grayscale phantoms

(b) Six colored phantoms

Figure 20: Six sample grayscale phantoms and colored phantoms used to train the mismatched distribution diffusion models

of $32 \times 32$ and $16 \times 16$. The probabilities of using these three patch sizes were $0.5, 0.3$, and $0.2$ respectively. For the whole image model, we kept all the parameters the same, but used a batch size of 8.

For image generation and inverse problem solving, we used a geometrically spaced descending noise level that was fine tuned to optimize the performance for each type of problem. We used the same set of parameters for the patch-based model and whole image model. The values without the self-supervised loss are as follows:

- CT with 20 and 60 views: $\sigma_{\max} = 10, \sigma_{\min} = 0.005$
- Deblurring: $\sigma_{\max} = 40, \sigma_{\min} = 0.005$
- Superresolution: $\sigma_{\max} = 40, \sigma_{\min} = 0.01$.

The values with the self-supervised loss are as follows:

- CT with 20 and 60 views: $\sigma_{\max} = 10, \sigma_{\min} = 0.01$
- Deblurring: $\sigma_{\max} = 1, \sigma_{\min} = 0.01$
- Superresolution: $\sigma_{\max} = 1, \sigma_{\min} = 0.01$.

Finally, for generating the CT images we used $\sigma_{\max} = 40, \sigma_{\min} = 0.005$.

When running Algorithm 1, we set $K = 10$ for all experiments and $M = 5$ for CT reconstruction and $M = 1$ for deblurring and superresolution. We ran 5 iterations of network backpropagation with a learning rate of $10^{-5}$. When using the LoRA module as in the ablation studies (see Tables 7 and 6), we ran 10 iterations of network backpropagation with a learning rate of $10^{-3}$.

The ADMM-TV method for linear inverse problems consists of solving the optimization problem

$$\operatorname{argmax}_{\boldsymbol{x}} \frac{1}{2} \|\boldsymbol{y} - A\boldsymbol{x}\|_2^2 + \lambda \operatorname{TV}(\boldsymbol{x}), \tag{13}$$

where TV($\boldsymbol{x}$) represents the L1 norm total variation of $vx$, and the problem is solved with the alternating direction method of multipliers. For CT reconstruction, deblurring, and superresolution, we chose $\lambda$ to be 0.001, 0.002, and 0.006 respectively.

The PnP-ADMM method consists of solving the intermediate optimization problem

$$\text{argmax}_{\boldsymbol{x}} f(\boldsymbol{x}) + (\rho/2)\|\boldsymbol{x} - (\boldsymbol{z} - \boldsymbol{u})\|_2^2, \tag{14}$$

where $\rho$ is a constant. The values for $\rho$ we used for CT reconstruction, deblurring, and super-resolution were 0.05, 0.1, and 0.1 respectively. We used BM3D as the denoiser with a parameter representing the noise level: this parameter was set to 0.02 for 60 view CT and 0.05 for the other inverse problems. A maximum of 50 iterations of conjugate gradient descent was run per outer loop. The entire algorithm was run for 100 outer iterations at maximum and the PSNR was observed to decrease by less than 0.005dB per iteration by the end.

The PnP-RED method consists of the update step

$$\boldsymbol{x} \leftarrow \boldsymbol{x} + \mu(\nabla f - \lambda(\boldsymbol{x} - D(\boldsymbol{x}))), \tag{15}$$

where $D(\boldsymbol{x})$ represents a denoiser. The stepsize $\mu$ was set to 0.01 for the CT experiments and 1 for deblurring and superresolution. We set $\lambda$ to 0.01 for the CT experiments and 0.2 for deblurring and superresolution. Finally, the denoiser was kept the same as the PnP-ADMM experiments with the same denoising strength.

Table 11 shows the average runtimes of each of the implemented methods when averaged across the test dataset for 60 view CT reconstruction.

Table 11: Average runtimes of different methods across images in the test dataset for 60 view CT recon.

| Method | Runtime (s) $\downarrow$ |
|---|---|
| Baseline | 0.1 |
| ADMM-TV | 1 |
| PnP-ADMM | 73 |
| PnP-RED | 121 |
| Whole diffusion | 112 |
| Whole SS | 248 |
| Whole LoRA | 329 |
| Patch diffusion | 123 |
| Patch SS | 289 |
| Patch LoRA | 377 |

## A.6 SELF-SUPERVISED INVERSE PROBLEM FIGURES

The following figures show additional examples of self-supervised inverse problem solving.

Figure 21 shows additional example slices of CT reconstruction from 60 views.

Figure 22 shows additional example slices of CT reconstruction from 20 views.

Figure 23 shows additional examples of deblurring with face images.

Figure 24 shows additional examples of superresolution with face images.

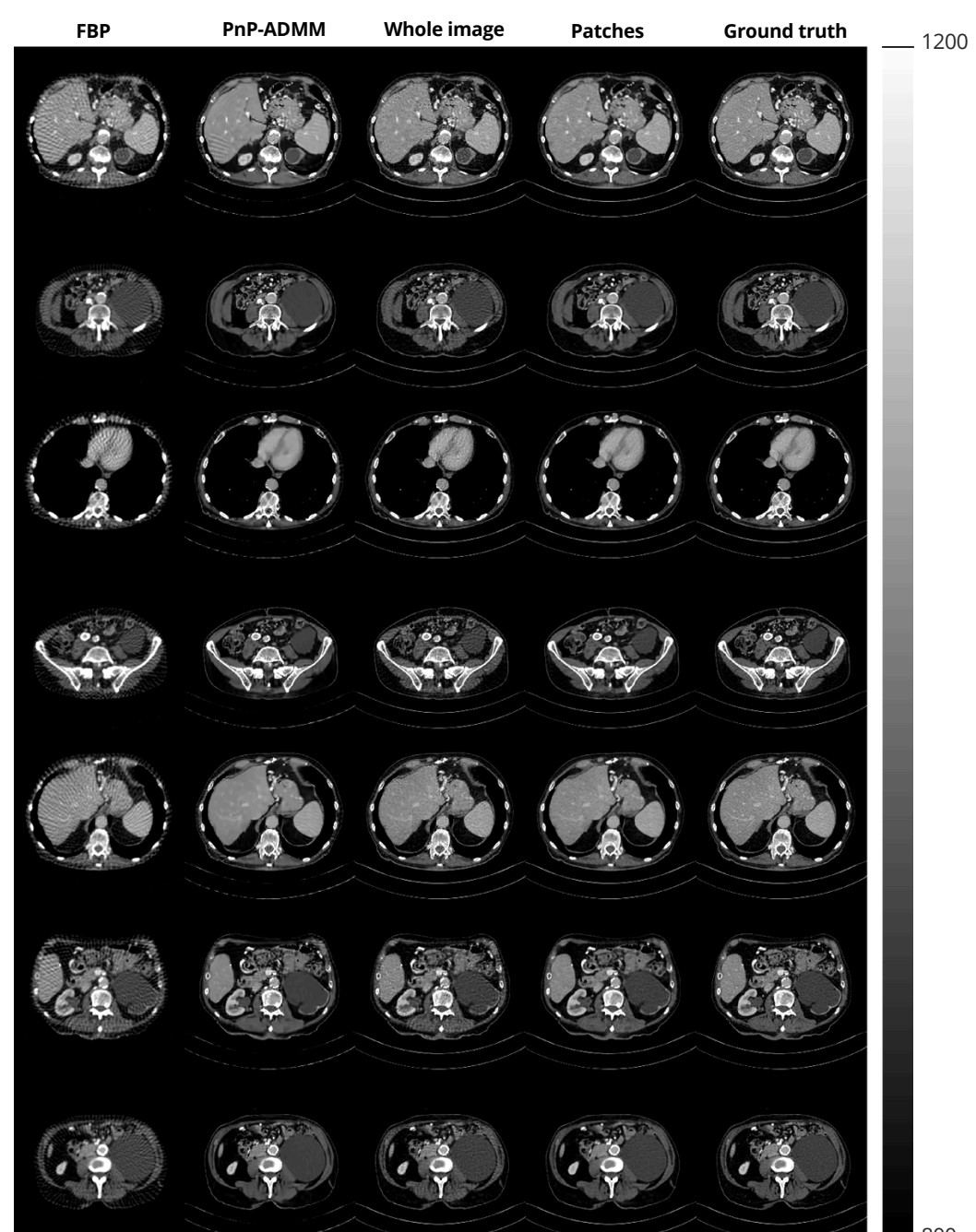

Figure 21: Additional figures for self-supervised 60 view CT recon.

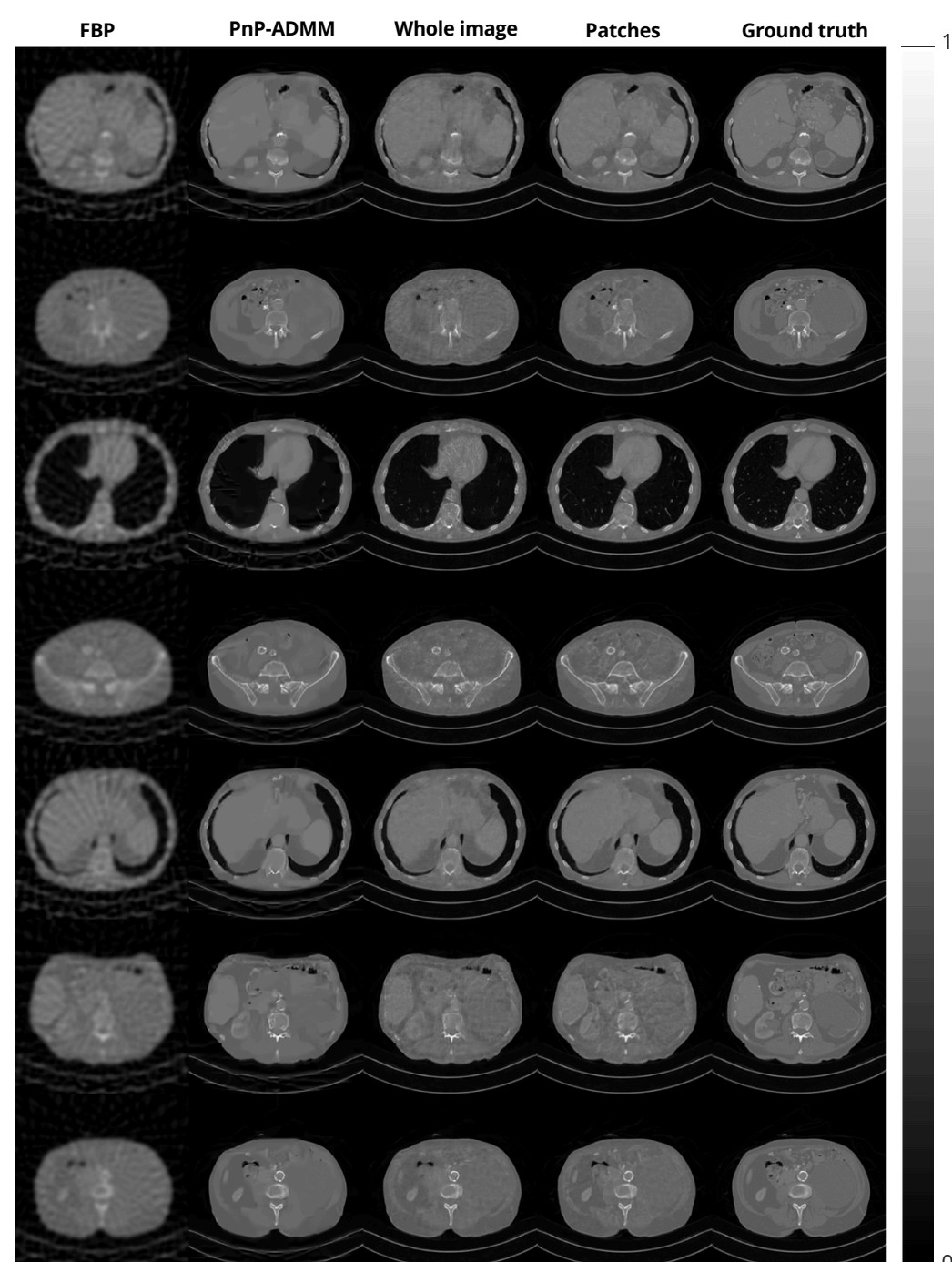

Figure 22: Additional figures for self-supervised 20 view CT recon.

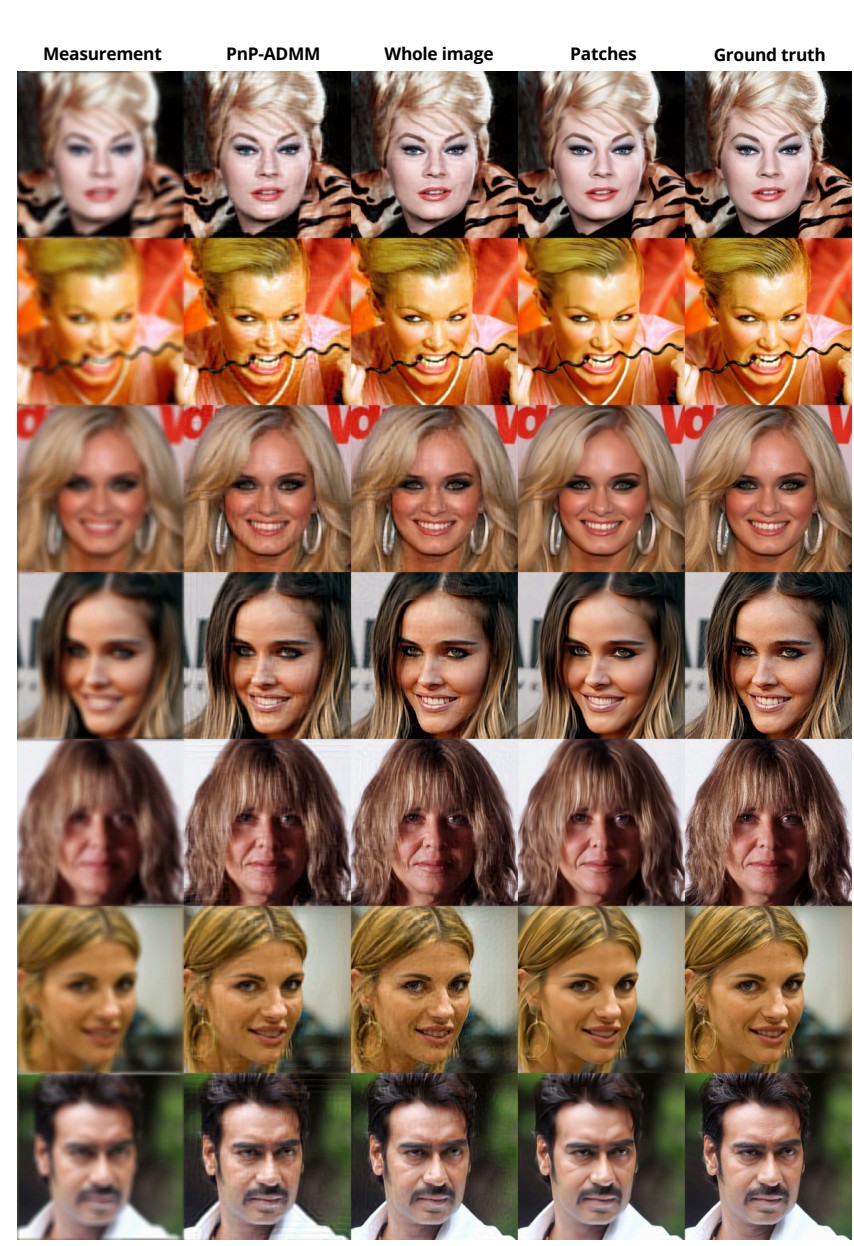

Figure 23: Additional figures for self-supervised deblurring.

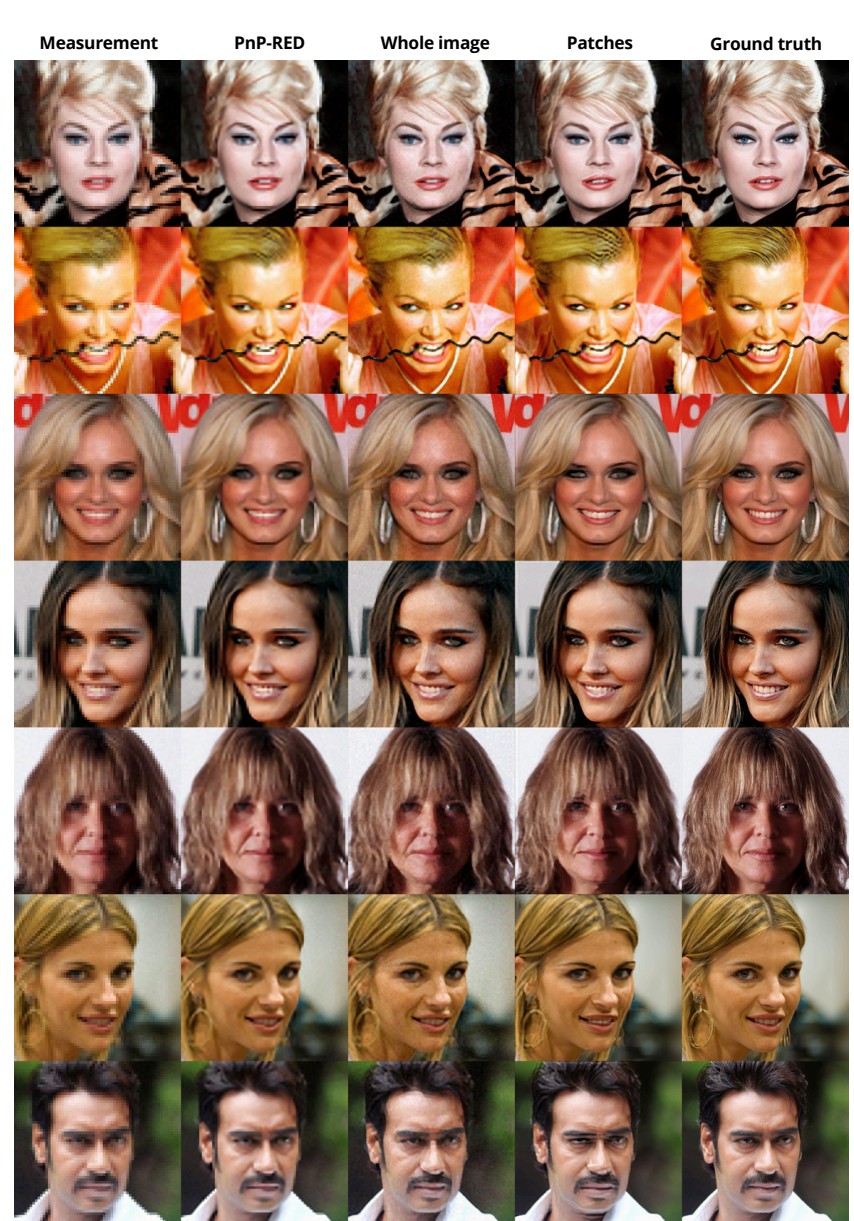

Figure 24: Additional figures for self-supervised superresolution.

### A.7 ADDITIONAL RESULTS

#### A.7.1 THEORETICAL ANALYSIS OF ALGORITHM 1.

This appendix provides a rough sketch of why Algorithm 1 should work better for the patch-based model compared to the whole-image model. Firstly, we rewrite (10) in the following form:

$$D(\boldsymbol{x}) = \sum_c \boldsymbol{G}'_c D_{\boldsymbol{\theta}}(\boldsymbol{G}_c \boldsymbol{x}, c), \tag{16}$$

where $c$ denotes the location of each patch, $\boldsymbol{G}_c$ denotes a patch extracting operator that extracts the patch corresponding to the location $c$ from the whole image $\boldsymbol{x}$, and thus $\boldsymbol{G}'_c$ is an operator that takes a patch and returns a whole image with the corresponding patch filled in (and the rest of the entries are zeros). Note that $c$ is input to the patch-based network $D_{\boldsymbol{\theta}}$ through positional encoding.

Now we analyze (11) using this framework. For simplicity we will first drop the conditional expectation and use the unconditional expectation; later we will analyze the conditional expectation. This gives us the loss function

$$L(\boldsymbol{\theta}) = \left\| \boldsymbol{y} - \boldsymbol{A} \sum_c \boldsymbol{G}'_c D_{\boldsymbol{\theta}}(\boldsymbol{G}_c \boldsymbol{x}, c) \right\|_2^2. \tag{17}$$

Next, we observe that when $\boldsymbol{A}$ is the blurring operator or the superresolution operator, $\boldsymbol{A}$ approximately operates independently on patches of a whole image. For example, particularly when the patches are not too small, blurring a whole image is approximately equal to blurring the individual patches and then assembling the patches together. This means that for every patch location $c$, there exists a patch operator $\boldsymbol{A}_c$ such that $\boldsymbol{G}_c \boldsymbol{A} \boldsymbol{x} \approx \boldsymbol{A}_c \boldsymbol{G}_c \boldsymbol{x}$, with $\boldsymbol{x}$ denoting the whole image. Likewise, $\boldsymbol{A} \boldsymbol{G}'_c \boldsymbol{x}_c \approx \boldsymbol{G}'_c \boldsymbol{A}_c \boldsymbol{x}_c$, where $\boldsymbol{x}_c$ denotes the image patch corresponding to the location $c$.

Hence we replace the loss in (17) with the approximately equal loss

$$L(\boldsymbol{\theta}) = \left\| \boldsymbol{y} - \sum_c \boldsymbol{A} \boldsymbol{G}'_c D_{\boldsymbol{\theta}}(\boldsymbol{G}_c \boldsymbol{x}, c) \right\|_2^2 \approx \left\| \boldsymbol{y} - \sum_c \boldsymbol{G}'_c \boldsymbol{A}_c D_{\boldsymbol{\theta}}(\boldsymbol{G}_c \boldsymbol{x}, c) \right\|_2^2. \tag{18}$$

Furthermore, since the patches in any given iteration of the reconstruction process are nonoverlapping, we have $\boldsymbol{y} = \sum_c \boldsymbol{G}'_c \boldsymbol{G}_c \boldsymbol{y}$. Thus the approximate loss is

$$L'(\boldsymbol{\theta}) = \left\| \sum_c \boldsymbol{G}'_c \boldsymbol{G}_c \boldsymbol{y} - \sum_c \boldsymbol{G}'_c \boldsymbol{A}_c D_{\boldsymbol{\theta}}(\boldsymbol{G}_c \boldsymbol{x}, c) \right\|_2^2 = \left\| \sum_c \boldsymbol{G}'_c (\boldsymbol{G}_c \boldsymbol{y} - \boldsymbol{A}_c D_{\boldsymbol{\theta}}(\boldsymbol{G}_c \boldsymbol{x}, c)) \right\|_2^2. \tag{19}$$

Now we have a sum of the form $\|X_1^2 + \ldots + X_n^2\|_2^2$, which Song et al. (2024) showed to be upper bounded by $K(\|X_1\|_2^2 + \ldots + \|X_n\|_2^2)$ for a fixed constant $K$. Applying this inequality gives

$$L'(\boldsymbol{\theta}) \leq K \sum_c \|\boldsymbol{G}'_c (\boldsymbol{G}_c \boldsymbol{y} - \boldsymbol{A}_c D_{\boldsymbol{\theta}}(\boldsymbol{G}_c \boldsymbol{x}, c))\|_2^2 = K \sum_c \|\boldsymbol{G}_c \boldsymbol{y} - \boldsymbol{A}_c D_{\boldsymbol{\theta}}(\boldsymbol{G}_c \boldsymbol{x}, c)\|_2^2. \tag{20}$$

Table 10 shows that performance is improved by using more backpropagation iterations for the loss function; hence, although in practice we only perform a fixed number of iterations for speed, optimally we should aim to reduce the loss $L(\theta)$ to zero. Based on (20), we can instead minimize the patchwise loss

$$L_c(\boldsymbol{\theta}) = \|\boldsymbol{G}_c \boldsymbol{y} - \boldsymbol{A}_c D_{\boldsymbol{\theta}}(\boldsymbol{G}_c \boldsymbol{x}, c)\|_2^2. \tag{21}$$

Observe that (21) has the same form as the loss that would be used for refining the network with a whole image model: $L_w(\boldsymbol{\theta}) = \|\boldsymbol{y} - \boldsymbol{A} D_{\boldsymbol{\theta}}(\boldsymbol{x})\|_2^2$. However, now instead of a loss of a single image, we now have individual losses of many patches of an image. For example, for the experiments of Table 1, 25 patches were used to tile each image for each diffusion iteration, so we would have 25 losses. This method of data augmentation provides an explanation for why the patch-based model obtains better performance than the whole-image model in the single measurement setting. We additionally note that although the positional encoding input into the network is different for each patch, the network does not separately learn a distribution for each position, as the weights are shared across these different positions. This is analogous to the analysis of Dhariwal & Nichol (2021), where a single diffusion model was trained on the 1000 classes of ImageNet with the class

label of the image being included as an additional input to the network. Since each class only had around 1000 images, it would have been very difficult to train a diffusion model on only one of the classes, but by training across all the classes at once, a much better network can be trained.

Next, we return to the conditional denoiser in (11). Recall that $D_{\boldsymbol{\theta}}(\boldsymbol{x}|\boldsymbol{y})$ is computed by first computing $D_{\boldsymbol{\theta}}(\boldsymbol{x})$ and then performing $C > 0$ iterations of conjugate gradient descent initialized at $D_{\boldsymbol{\theta}}(\boldsymbol{x})$. For simplicity, here we analyze the effect of performing gradient descent on $\|\boldsymbol{y} - \boldsymbol{A}\boldsymbol{x}\|_2^2$.

From before, we have $\boldsymbol{y} - \boldsymbol{A}\boldsymbol{x} = \sum_c \boldsymbol{G}_c'(\boldsymbol{G}_c\boldsymbol{y} - \boldsymbol{A}_c\boldsymbol{G}_c\boldsymbol{x})$. Making the same assumptions as before that the patches are nonoverlapping and $\boldsymbol{A}$ operates on patches roughly independently, we have

$$\boldsymbol{A}'\boldsymbol{y} = \boldsymbol{A}' \sum_c \boldsymbol{G}_c'\boldsymbol{G}_c\boldsymbol{y} = \sum_c \boldsymbol{A}'\boldsymbol{G}_c'\boldsymbol{G}_c\boldsymbol{y} \approx \sum_c \boldsymbol{G}_c'\boldsymbol{A}_c'\boldsymbol{G}_c\boldsymbol{y} \tag{22}$$

Hence, using $c$ and $d$ as patch location indices, the gradient is

$$\boldsymbol{g} = \boldsymbol{A}'(\boldsymbol{y} - \boldsymbol{A}\boldsymbol{x}) = \sum_c \boldsymbol{G}_c'\boldsymbol{A}_c'\boldsymbol{G}_c \sum_d \boldsymbol{G}_d'(\boldsymbol{G}_d\boldsymbol{y} - \boldsymbol{A}_d\boldsymbol{G}_d\boldsymbol{x}). \tag{23}$$

Since the patches are nonoverlapping, $\boldsymbol{G}_c\boldsymbol{G}_d' = 0$ unless $c = d$, in which case it is the identity. Hence, we can combine the double sum into a single sum to get

$$\boldsymbol{g} = \sum_c \boldsymbol{G}_c'\boldsymbol{A}_c'(\boldsymbol{G}_c\boldsymbol{y} - \boldsymbol{A}_c\boldsymbol{G}_c\boldsymbol{x}). \tag{24}$$

Note that this takes a very similar form to the full image gradient $\boldsymbol{g} = \boldsymbol{A}'(\boldsymbol{y} - \boldsymbol{A}\boldsymbol{x})$: in particular, we are simply replacing the full images $\boldsymbol{x}$ and $\boldsymbol{y}$ with patches $\boldsymbol{G}_c\boldsymbol{x}$ and $\boldsymbol{G}_c\boldsymbol{y}$ and then summing over all the patches. Thus, $D_{\boldsymbol{\theta}}(\boldsymbol{x}|\boldsymbol{y})$ acts independently on the patches of $D_{\boldsymbol{\theta}}(\boldsymbol{x})$. Finally, it is readily shown that when using $D_{\boldsymbol{\theta}}(\boldsymbol{x}|\boldsymbol{y})$ in place of $D_{\boldsymbol{\theta}}(\boldsymbol{x})$ for (21), we again get a sum of losses over all patches.

Next, we turn to the case of CT reconstruction where $\boldsymbol{A}$ does not operate approximately independently on patches. Here, we consider the network refining process in Algorithm 1 during the final diffusion iteration. By treating the entire conditional denoiser $D_{\boldsymbol{\theta}}(\boldsymbol{x}|\boldsymbol{y})$ as parametrized by one network $f_{\boldsymbol{\theta}}(\boldsymbol{x})$, the loss function simply becomes $L(\boldsymbol{\theta}) = \|\boldsymbol{y} - \boldsymbol{A}f_{\boldsymbol{\theta}}(\boldsymbol{x})\|_2^2$. This is equivalent to the standard DIP formulation. Baguer et al. (2020) proved that for this formulation, provided that certain technical conditions hold, the minimization problem for $\theta$ will converge when gradient descent is applied. Most importantly, we need the following definition:

**Definition:** An activation function $\sigma : \mathbb{R}^n \to \mathbb{R}^n$ is *valid* if it is continuous, monotone, and bounded, i.e., there exists $c > 0$ such that $\forall x, \|\sigma(x)\| \leq c\|x\|$.

For the UNets we used to train the diffusion models, the activation functions were ReLUs and sigmoid linear unit (SiLU) which are defined as $\text{SiLU}(x) = x \cdot \sigma(x)$, where $\sigma(x)$ is the sigmoid function. Both of these are valid, so the network refining process will converge at the final diffusion iteration.

### A.7.2 ADDITIONAL EXPERIMENTS

Tables 12 and 13 show LPIPS perception scores for various methods in the single measurement setting and small dataset setting, respectively. We used the VGG network (Zhang et al., 2018) to compute these scores and averaged them across all the images in the test dataset. These results show that our proposed method obtained the images with the best visual image quality.

Table 12: LPIPS score for different methods in single measurement setting.

| Method | Deblur | Superresolution |
|---|---|---|
| ADMM-TV | 0.469 | 0.542 |
| PNP-ADMM | 0.316 | 0.701 |
| PNP-RED | 0.331 | 0.303 |
| Whole image, naive | 0.469 | 0.469 |
| Patches, naive | 0.440 | 0.465 |
| Self-supervised, whole | 0.275 | 0.339 |
| Self-supervised, patch (Ours) | **0.238** | **0.264** |
| Whole image, correct* | 0.194 | 0.249 |
| Patches, correct* | 0.222 | 0.244 |

*not available in practice for mismatched distribution inverse problems

Table 13: LPIPS score for different methods in small dataset setting.

| Method | Deblur | Superresolution |
|---|---|---|
| Best baseline | 0.316 | 0.303 |
| Whole | 0.23 | 0.289 |
| Patch | **0.219** | **0.259** |

Algorithm 2 provides the pseudocode for the "whole image, correct" method of Table 1.

---

**Algorithm 2** Whole image recon, no distribution shift

---

**Require:** $\sigma_1 < \sigma_2 < \ldots < \sigma_T$, $\epsilon > 0$, $C > 0$, $\boldsymbol{y}$
  Initialize $\boldsymbol{x} \sim \mathcal{N}(0, \sigma_T^2 \boldsymbol{I})$
  **for** $i = T : 1$ **do**
    Sample $z \sim \mathcal{N}(0, \sigma_i^2 \boldsymbol{I})$
    Set $\alpha_i = \epsilon \cdot \sigma_i^2$
    Apply neural network to get $D = D_\theta(\boldsymbol{x}, \sigma_i)$
    Run $C$ iterations of CG for (7) initialized with $D$
    Set $\boldsymbol{s} = (D - \boldsymbol{x})/\sigma_i^2$
    Set $\boldsymbol{x}$ to $\boldsymbol{x} + \frac{\alpha_i}{2}\boldsymbol{s} + \sqrt{\alpha_i}\boldsymbol{z}$
  **end for**
Return $\boldsymbol{x}$.

---

To demonstrate more statistically significant results, we ran the inverse problems of Table 1 over test datasets of 25 images in the single measurement setting. Table 14 shows these results: our method using patches consistently outperforms the baseline and using the whole image model.

Figures 25 and 26 compare the PSNR of the whole-image model and our proposed patch-based model for each individual image in the 25 image test dataset. The plots show that for all but one test image in the CT case and for all test images in the deblurring case that the patch-based model outperformed the whole-image model, illustrating the consistency of our method.

Table 14: Extended inverse problem solving results using 25 image test dataset in single measurement setting. Best results are in bold. A two sample t-test shows that for all the experiments, the PSNR and SSIM of the patch-based model exceeds that of the whole image model and the result is statistically significant, all with p-value less than $10^{-7}$.

| Method | CT, 20 Views | | CT, 60 Views | | Deblurring | | Superresolution | |
|---|---|---|---|---|---|---|---|---|
| | PSNR↑ | SSIM↑ | PSNR↑ | SSIM↑ | PSNR↑ | SSIM↑ | PSNR↑ | SSIM↑ |
| PnP-ADMM | 30.44 | 0.843 | 37.02 | 0.937 | 29.93 | 0.835 | 26.37 | 0.771 |
| Whole image, SS | 33.51 | 0.866 | 40.61 | 0.959 | 30.19 | 0.839 | 27.78 | 0.715 |
| Patches, SS (Ours) | **34.59** | **0.891** | **41.55** | **0.968** | **30.95** | **0.867** | **28.68** | **0.839** |

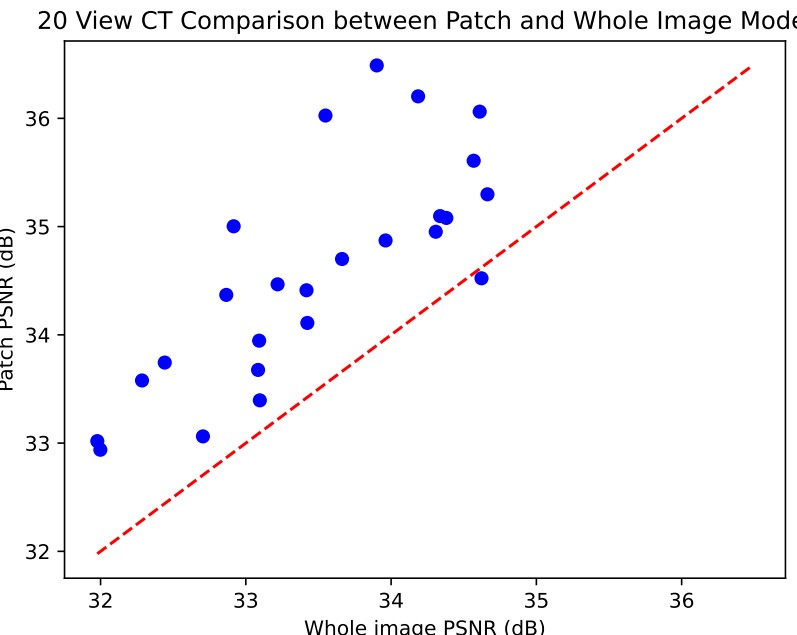

Figure 25: Comparison between PSNR of 20 view CT reconstruction between whole-image model and patch-based model for each image in the test dataset.

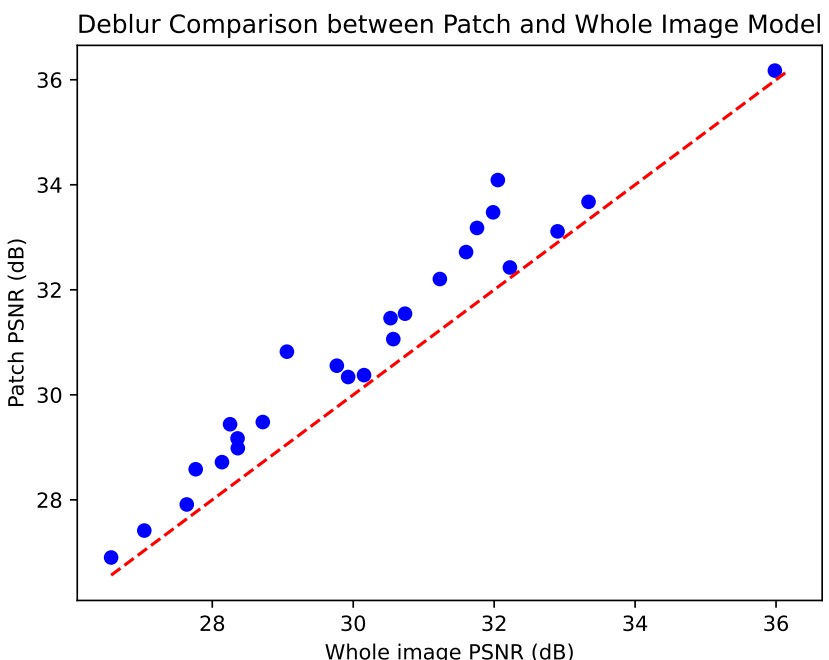

Figure 26: Comparison between PSNR of deblurring between whole-image model and patch-based model for each image in the test dataset.

### A.7.3 EXPERIMENTS ON LARGER IMAGES

To show that our method scales to larger images, we ran experiments on 60 view CT reconstruction and deblurring with $512 \times 512$ images. For the CT experiments, we still used the AAPM dataset (McCollough et al., 2017) processed in the same way as for Table 1, but kept the slices in their original size of $512 \times 512$. For deblurring, we used the FFHQ dataset (Karras et al., 2019) which contains images of size $512 \times 512$. We scaled each of the RGB channels to between 0 and 1. We used a uniform blur kernel of size $17 \times 17$ and added noise with $\sigma = 0.01$. We used the same patch-based networks trained for Table 1 as initializations for these out of distribution experiments. Table 15 shows results of these experiments, where our method obtained the highest quality reconstruction. Although the improvement is modest, note that we trained the patch-based model on phantoms images of size $256 \times 256$, which is extremely far out of distribution from the test dataset.

Table 15: Results of inverse problem solving in single measurement setting for $512 \times 512$ images.

| Method | CT, 60 views | | Deblur | |
|---|---|---|---|---|
| | PSNR↑ | SSIM ↑ | PSNR↑ | SSIM ↑ |
| Baseline | 28.33 | 0.700 | 24.11 | 0.649 |
| ADMM-TV | 29.36 | 0.788 | 28.14 | 0.760 |
| PnP-ADMM | 37.48 | 0.910 | 29.77 | 0.812 |
| Patch, naive | 29.32 | 0.793 | 26.58 | 0.749 |
| Patch, SS | **37.82** | **0.919** | **30.35** | **0.825** |

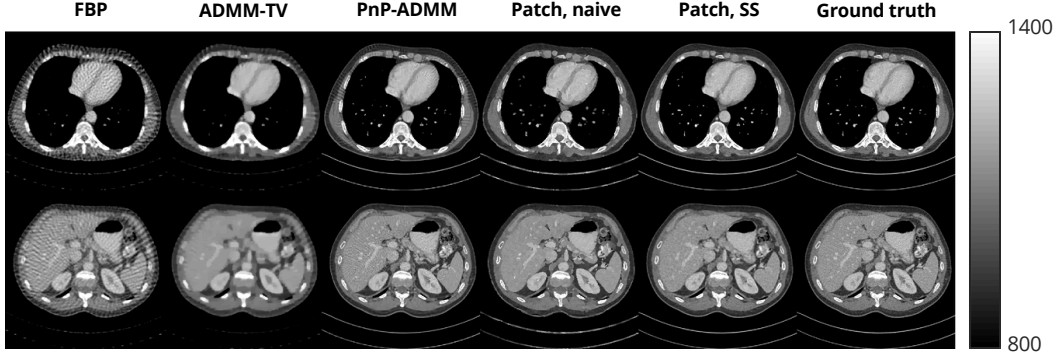

Figure 27: Results of 60 view reconstruction on $512 \times 512$ images in single measurement setting displayed in Hounsfield units.

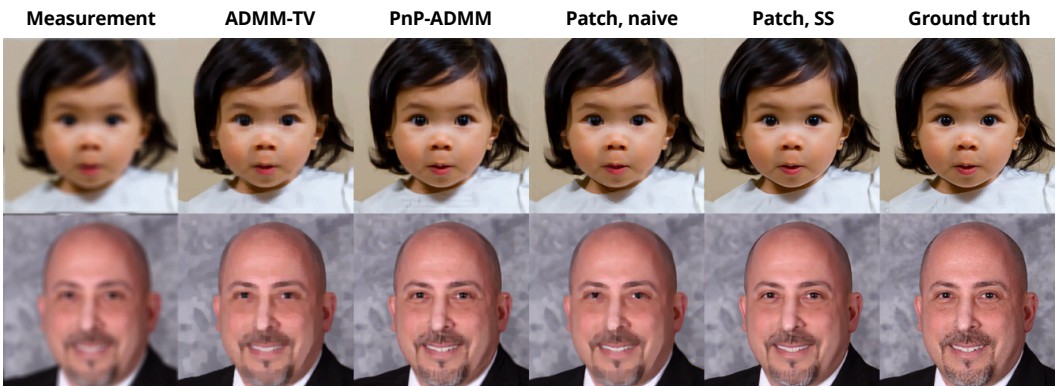

Figure 28: Results of deblurring on $512 \times 512$ images in single measurement setting.

### A.7.4 ALTERNATE DIFFUSION INVERSE SOLVERS

We ran comparison experiments between various state-of-the-art diffusion inverse solving algorithms including Chung et al. (2024), Wang et al. (2022), and Chung et al. (2023a). Note that Chung et al. (2024) developed a method targeting 3D inverse problems, but used conjugate gradient descent to enforce data consistency. Hence, that method closely resembles the methods we used in our main work (see Tables 1 and 2). Additionally, note that these three methods assume that we have a network trained on in-distribution data and thus do not refine the network on the fly. Hence, when applying them in the single measurement setting, we used the network that was trained on the ellipse phantoms and did not refine the network according to the measurement.

Tables 16 and 17 show the results of using these methods in the single measurement and small dataset settings, respectively. In the single measurement setting, since these methods do not account for the mismatched prior, the results are very poor compared to our method that adjusts for the mismatched prior. Figures 29 and 30 show the failures of these methods visually. Since the network was trained on ellipse phantoms, the reconstructed images exhibit excessively smooth and rounded features.

Table 16: Comparison between different diffusion inverse solving methods in single measurement setting.

| Method | CT, 60 views | | Deblur | |
|---|---|---|---|---|
| | PSNR↑ | SSIM↑ | PSNR↑ | SSIM↑ |
| DPS (Chung et al., 2023a) | 28.28 | 0.729 | 27.54 | 0.864 |
| DDNM (Wang et al., 2022) | 23.21 | 0.572 | 24.21 | 0.761 |
| DDS (Chung et al., 2024) | 33.97 | 0.934 | 26.77 | 0.782 |
| Ours | **41.70** | **0.967** | **30.34** | **0.860** |

Table 17: Comparison between different diffusion inverse solving methods in small dataset setting.

| Method | CT, 60 views | | Superresolution | |
|---|---|---|---|---|
| | PSNR↑ | SSIM↑ | PSNR↑ | SSIM↑ |
| DPS (Chung et al., 2023a) | 34.24 | 0.828 | 28.15 | 0.814 |
| DDNM (Wang et al., 2022) | 26.86 | 0.860 | 28.08 | 0.816 |
| DDS (Chung et al., 2024) | **41.21** | **0.965** | **28.28** | **0.830** |

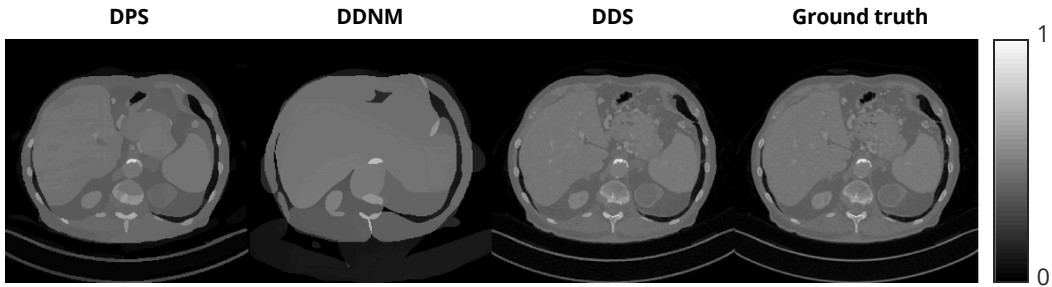

Figure 29: Visual results of 60 view CT reconstruction using different diffusion inverse solvers in single measurement setting.

| DPS | DDNM | DDS | Ground truth |

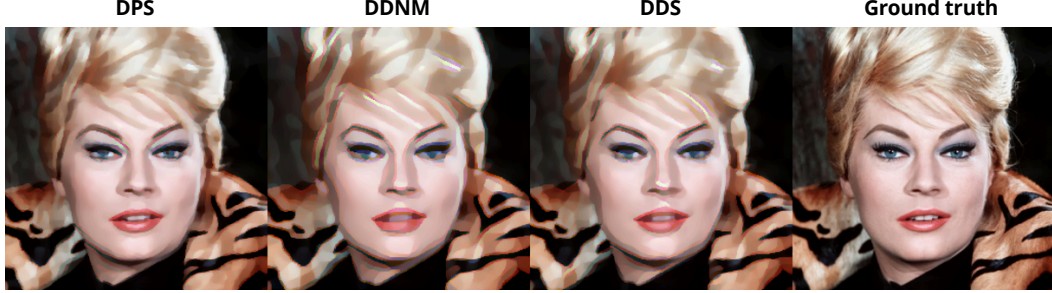

Figure 30: Visual results of deblurring using different diffusion inverse solvers in small dataset setting.

### A.7.5 FURTHER BREAKDOWN OF METHODS

It may seem surprising that in Table 2, where a small sample of training data is available, the results from using the patch-based model are generally worse than those of Table 1, where only a single measurement is available. This is because for Table 2, after we have fine-tuned the network using the small dataset, we assume that the network is now in-distribution to the test data, so no network refining is performed when solving the inverse problem. In contrast, Algorithm 1 is used for Table 1 to refine the network according to the measurement. We illustrate this point further by running experiments where we applied this network refinement method even after the network was fine-tuned using the small dataset. Table 18 shows that significant improvements can be made from this further refinement to fit the measurement. We see that regardless of whether the network was fine-tuned (FT) with the small dataset, including the self-supervision network refining step results in better performance.

Table 18: Inverse problem solving results using self-supervision (Algorithm 1) where the initial network was obtaining by fine-tuning on the 10 image dataset.

| Method | CT, 20 Views | | CT, 60 Views | | Deblurring | | Superresolution | |
|--------|--------------|---------------|--------------|---------------|------------|---------------|-----------------|---------------|
| | PSNR↑ | SSIM↑ | PSNR↑ | SSIM↑ | PSNR↑ | SSIM↑ | PSNR↑ | SSIM↑ |
| Naive | 28.11 | 0.800 | 33.10 | 0.911 | 25.85 | 0.742 | 25.65 | 0.742 |
| SS only | 33.77 | 0.874 | 41.45 | 0.966 | 30.34 | 0.860 | 28.10 | 0.827 |
| FT only | 33.44 | 0.875 | 41.21 | 0.965 | 29.25 | 0.840 | 28.28 | 0.830 |
| FT and SS | **36.43** | **0.914** | **42.42** | **0.971** | **30.56** | **0.867** | **28.38** | **0.831** |

We can quantify the degree to which the networks are memorizing the training datasets. We generated 100 images using both the patch-based model and the whole-image model after fine-tuning with the small dataset of CT images. For each generation, we compared the NRMSD (normalized root mean square difference) of the image with each of the 10 CT training images to find the one with the smallest NRMSD. This was computed by the formula

$$\text{NRMSD}(x, y) = \frac{1}{x_{\max} - x_{\min}} \sqrt{\frac{\sum_{i=1}^{n}(x_i - y_i)^2}{n}}, \tag{25}$$

where $x$ is the image in the dataset, $y$ is the generated image, and $n$ is the number of pixels in the image. Table 19 shows the number of images from the 100 generated images where the smallest NRMSD was less than a certain threshold. This illustrates that the whole image model tended to memorize the training dataset much more than the patch-based model.

Table 19: Images of the 100 generations that had lowest NRMSD with one of the images in the small dataset below a certain threshold.

| NRMSD (%) | Whole Image | Patches |
|-----------|-------------|---------|
| 5 | 37 | 0 |
| 10 | 85 | 4 |

### A.7.6 ADDITIONAL TRAINING IMAGES

Figures 31 and 32 show some more samples of the ellipse phantoms that were used to train the diffusion models.

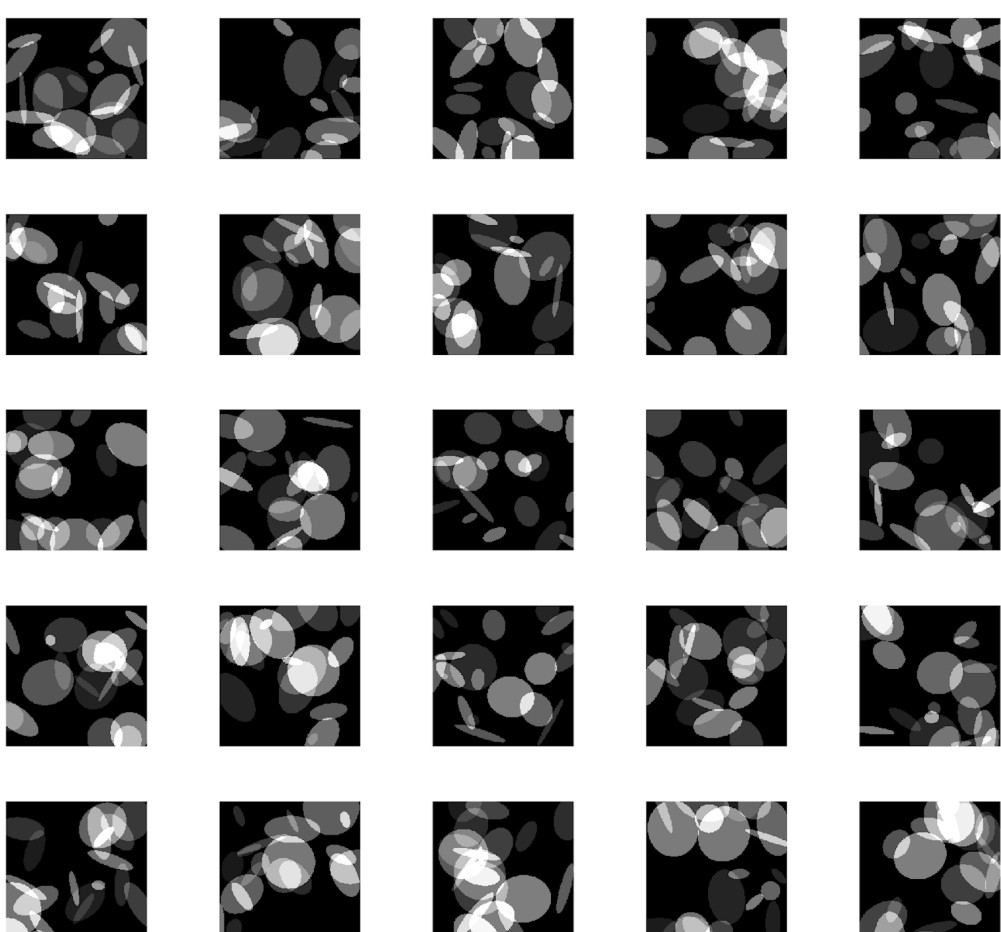

Figure 31: 25 sample grayscale phantoms used to train the mismatched distribution diffusion models.

2052
2053
2054
2055
2056
2057
2058
2059
2060
2061
2062
2063
2064
2065
2066
2067
2068
2069
2070
2071
2072
2073
2074
2075
2076
2077
2078
2079
2080
2081
2082
2083
2084
2085
2086
2087
2088
2089
2090
2091
2092
2093
2094
2095
2096
2097
2098
2099
2100
2101
2102
2103
2104
2105

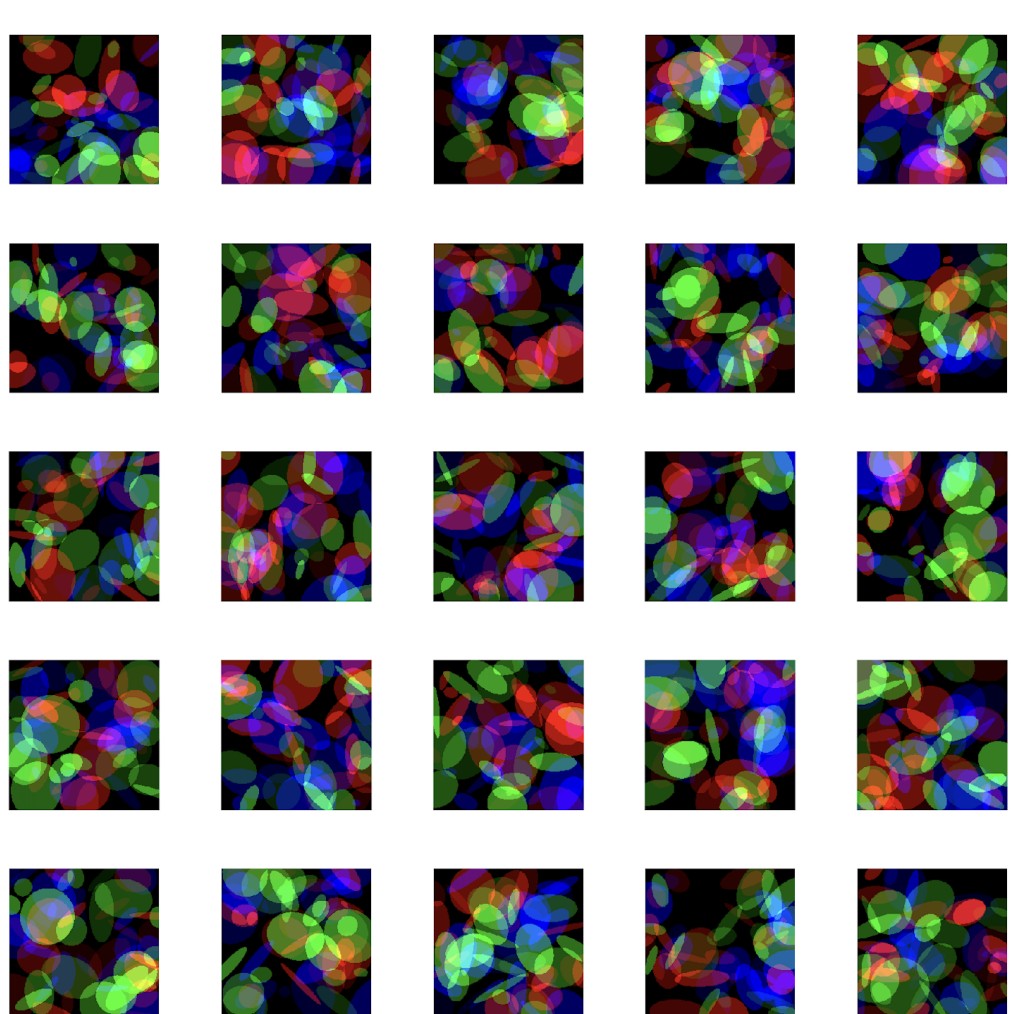

Figure 32: 25 sample color phantoms used to train the mismatched distribution diffusion models.

