# OpenReview forum: "Patch-Based Diffusion Models Beat Whole-Image Models for Mismatched Distribution Inverse Problems"
_ICLR.cc/2025/Conference — ICLR 2025 Conference Withdrawn Submission_

### Official Review · Reviewer_Quvp · 2024-10-21

**Soundness:** 2
**Presentation:** 1
**Contribution:** 3
**Rating:** 5
**Confidence:** 4

**Summary:**

This works tackles the challenging problem of solving inverse problems where only a few samples are available from the test distribution. Authors propose a patch-based diffusion prior in this scenario, that learns the image distribution from patches, and not from whole images. Authors argue that whole-image models are prone to overfitting to the data distribution, and thus are unable to provide sufficient performance when the test samples come from out of distribution. Numerical experiments are provided to verify these findings.

**Strengths:**

- The problem is well motivated and important. In a multitude of real-world applications, it is infeasible to collect large datasets for training diffusion models on the target distribution. However, finding ways to leverage the strong prior provided by diffusion models to tackle such problems is a valuable direction.

- The key idea of using patch-based diffusion instead of whole-image diffusion is intuitive and sensible. The patch-based model is less prone to overfitting to the available source data distribution, and thus can perform better in the presence of distribution shifts. Moreover, patch-based models can be trained more data-efficiently which is crucial in the data scarce domain.

- The experimental results, if verified on larger scale experiments, are promising.

**Weaknesses:**

- The clarity of the paper could be greatly improved. In particular, I had a difficult time following 3.1, which is central to understanding the proposed algorithm and led to downstream confusions throughout the paper. Specific questions follow under 'Questions'.

- If I understand correctly, the experimental results are reported on 10 samples. This is not enough to report statistically meaningful results.

- Many claims in the experimental section are vague or not supported properly by the experiments (see more details under 'Questions').

**Questions:**

Questions/feedback on clarity:

- 3.1. is difficult to follow without already knowing the framework the authors adapt. Why is the bordering region added? What is $M$ here, and what is $k$? What does $i$ denote? Is $x$ an image patch or the whole image, as it has been used throughout the paper for both. Why only the x-positions of patches are concatenated as input, why not both x and y coordinates?

- Probably due to the unclear nature of 3.1. I was unable to properly follow 3.2 in some parts. What do authors mean by "the outermost product is computationally very expensive"?

Questions/feedback on experiments:

- How many samples have been used to produce the results reported in Table 1?

- I recommend reporting perceptual metrics such as LPIPS as well, especially for image deblurring and superresolution.

- How is it possible that the proposed method, without training, outperforms diffusion approaches that leverage training data? This sounds very counter-intuitive.

- Which dataset has been used to produce Figures 4 and 5?

- The discussion on diversity of generated samples is very vague. What do authors mean by samples "show some unrealistic features"? IT is unclear based on the Figure 7 which features are considered realistic/unrealistic. Claims about sample diversity would be more convincing if authors reported specific metrics about diversity, such as Recall.

- Table 3 is in the appendix, and therefore it should either moved to the main paper or the discussion about Table 3 should be moved to the appendix.

---

> ### Author Response · Authors · 2024-11-21
> **Response to reviewer Quvp**
>
> We thank the reviewer for their valuable feedback and insights.
>
> **Comment: Experimental results on 10 figures are not enough to report statistically meaningful results**
>
> Response: In Table 14 we added comparisons across a test dataset of 25 images. Additionally, we added Figures 25 and 26 to the paper that show comparisons between the PSNR of the whole image model and our proposed patch-based model for each individual image in the test dataset. The plots show that for all but one test image in the CT case and for all test images in the deblurring case that the patch-based model outperforms the whole-image model, illustrating the consistency of our method.
>
> **Comment: 3.1 is difficult to follow**
>
> Response: The bordering region is added so that there are many possible ways to tile the central image with patches. If we were to use the same method of tiling the central image with patches for every diffusion iteration during the reconstruction process, there would be boundary artifacts between the patches. By varying the random patch location in Figure 1 for each diffusion iteration, we change the tiling method and eliminate boundary artifacts.
>
> There was duplicate notation used in the original submission; originally, in Section 3.1, $M^2$ was stated to be the number of ways to choose the random patch location in Figure 1 (so that $M$ is the number of ways to choose the horizontal and vertical locations), and $M$ was also set to be the number of conjugate gradient iterations to be run in Section 3.2 and Algorithm 1. We have fixed the second use of $M$ so that now, $C$ represents the number of conjugate gradient iterations. Also, $k$ is the floor of $N/P$, so that $k+1$ patches in both directions are needed to fully cover the central image, hence the sum in eq. (8) ranges from 1 to $(k+1)^2$. In eq. (8), $i$ denotes an integer that indexes all the possible ways to select a random patch offset.
>
> In eq. (8), the $x$ on the LHS denotes the whole image with zero padding, as we are modeling the distribution of the entire image. However, in eq. (9), the $x$ denotes extracting a random patch from an image in the training dataset. In all subsequent places where $x_t$ occurs in Section 3.2, it refers to the whole image. Finally, both the $x$ and $y$ coordinates of the patches are concatenated as inputs to the network. We modified the corresponding statement in the paper to make this more clear. We have also provided additional explanations in Section 3.1 to make the algorithm and probability model more clear.
>
> **Comment: What do the authors mean by the outermost product is computationally very expensive**
>
> Response: The outermost product refers to the outer product that iterates from $i=1$ to $M^2$ in eq. (8). When using this form of the distribution to compute the score function, the log in the score turns the product into a sum with $M^2$ terms. Since $M=64$ for most of our experiments, this would involve computing a sum over 4096 terms. As this computation would need to be performed for each of the 1000 diffusion iterations, this would be undesirably slow. Thus, instead of computing all 4096 terms and summing over them, at each diffusion iteration we randomly choose one of the terms and compute only this term.
>
> **Comment: How many samples were used in Table 1?**
>
> Response: We used average metrics over a test dataset of 10 images in Table 1. We now provide average metrics over a larger test dataset of 25 images in Table 14.
>
> **Comment: I recommend reporting perceptual metrics such as LPIPS**
>
> Response: We added Tables 12 and 13 to the revised paper which show the LPIPS scores for the deblurring and superresolution experiments in both the single measurement and small dataset settings. These results show that our proposed method (with the same parameters as in all previous results) obtains the images with the best visual image quality. Appendix A.7.2 provides further details.

---

> ### Author Response · Authors · 2024-11-21
> **Response to reviewer Quvp (part 2)**
>
> **Comment: How is it possible that the proposed method without training outperforms diffusion approaches that leverages training data**
>
> Response: It is indeed true that in Table 2, where a small sample of training data is available, the results from using the patch-based model are generally worse than those of Table 1 where only a single measurement is available. The main reason for this is that after fine-tuning a diffusion model using a small dataset of in-distribution data, the inverse solver assumes that the model has learned the correct prior for the test images. Therefore, when using this trained network to solve inverse problems, we simply use an existing diffusion inverse solving method without further refining the network through self-supervised learning based on the measurements data. On the other hand, the proposed method in the single measurement setting first takes a network that is known to have learned a prior that does not match the images in the test dataset, so we perform on the fly network refining via Algorithm 1. This difference in algorithm is likely to have contributed to this performance discrepancy in some tasks.
>
> We further explored this point through the new Table 18. We conducted additional experiments by first fine-tuning (FT) the network on the small dataset and then applying the self-supervised learning (SS) based on measurements (Algorithm 1). The results show that this hybrid method (FT+SS) significantly improved performance over the FT-only or SS-only methods. We also note that the second best method may be either FT-only or SS-only method depending on the specific task. Hence, since the performance gap between training data (ellipse phantom images) and testing data (CT images or facial images) is significant, with only 10 “in-distribution” data samples to fine-tune the network, it is hard to adapt to the correct prior for testing domain. We conjecture this is because the distribution of CT images or facial images in question is very high dimensional and may contain many modes. Hence, refining it to fit a specific measurement can still have significant benefits.
>
> **Comment: Which dataset has been used for Figures 4 and 5?**
>
> Response: Figures 4 and 5 are the results of CT reconstruction with 60 views, so we used the AAPM 2016 CT challenge data. We provided more detailed information about this dataset in the experimental setup part of Section 4, and provided additional clarifications in the revised paper.
>
> **Comment: Discussion on diversity of generated samples is very vague**
>
> Response: Since the dataset used to fine-tune the networks is very small (10 images), it is difficult to meaningfully compute quality-of-sample metrics such as FID (Frechet inception distance) or IS (inception score) or diversity metrics. However, we can quantify the degree to which the networks are memorizing the training datasets. We generated 100 images using both the patch-based model and the whole image model after fine-tuning with the small dataset of CT images. For each generation, we compared the NRMSD (normalized root mean square difference) of the image with each of the 10 CT training images to find the one with the smallest NRMSD. We now summarize these results in the new Table 19, which illustrates that the whole-image model tended to memorize the training dataset much more than the patch-based model.
>
> **Comment: Table 3 is in the appendix and should be moved**
>
> Response: Table 3 provides the same information as Figures 4 and 5 but in a table format. The message of Figures 4 and 5 is that when fine-tuning the whole image model for an excessively long duration of time, it will overfit to the data and obtain worse image reconstructions, but patch-based models can avoid this issue. This is easily seen from Figures 4 and 5 from the trend of the curves, but is less clear when presented in the table format of Table 3. Hence, to avoid redundancy of information, in light of the page limit, we put Table 3 in the appendix.

---

> > ### Comment · Reviewer_Quvp · 2024-11-26
> > **Response to authors**
> >
> > Thank you for the response.
> >
> > The discussion in 3.1. is still confusing. In fact, I had to carefully read the framework in [1] to understand what equation (8) represents. In particular, there is $M$ and $k$ in equation (8) which is not defined before. The outer product should go up to $P^2$ if I follow the notation of this paper correctly (which is doubly confusing because [1] used $P$ for patch size), as $P$ is defined as padding size. However, later the patch size is referred to as $P \times P$. Also, the patch offset should have two indices, one for row and one for column. Overall, I would recommend a more thorough and precise introduction of the framework.
> >
> > On the evaluation set size: Can the authors justify why such a low number (10 or 25) of images is used for evaluations? Especially in the general domain experiments for superresolution and deblurring, one could have access to thousands of images for evaluation to make a more convincing argument.
> >
> > [1] Learning Image Priors through Patch-based Diffusion Models for Solving Inverse Problems. Jason Hu, et al.

---

> ### Author Response · Authors · 2024-11-26
> **Response to reviewer**
>
> We thank the reviewer for reading our response and providing additional feedback. We further rephrased much of section 3.1 to make the notation more clear. In particular, while [1] used $P$ for the patch size and $M$ for the padding, we found that it was sufficient to consider the model where the padding and patch size were equal (both set to 64 for the main experiments), so we used $P$ to denote both of these quantities. We have removed references to $M$ in the revision as this should be equal to $P$ in our model and also restated the definition of $k$. Finally, for the batch offset, we added the clarification that each offset should indeed be specified by two indices. However, in eq. (8) and onwards in the paper, we used one index $i$ to denote the patch offset so that it would be consistent with the single index $r$ (also in eq. (8)) which corresponds to the particular $P \times P$ patch. (For example, the index 1 could represent the offset (0,0), 1 could represent the offset (0,1), etc.) It would have been possible to use two indices for both the patch and the offset, but this would have led to an excessive amount of notation and subscripts. We also updated Figure 1 to make this more clear.
>
> We believe that our experiments on test datasets of 25 images sufficiently demonstrates the superiority of using the patch-based model for the following reasons. Firstly, the test datasets were randomly drawn from the entire AAPM and CelebA datasets and are representative of their respective datasets. Secondly, to confirm that the results we obtained for these models are statistically significant, we ran two sample t-tests comparing the results from using the patch-based model and whole image model. In particular, using the 25 image test dataset experiments shown in Table 14, we compared the sample PSNR obtained by the patch-based model versus the whole image model for each of the four inverse problems. In each case, a two sample t-test found that the mean PSNR when using patches was higher than the mean PSNR when using the whole image, with a p-value less than $10^{-7}$. We repeated this test for the SSIM in Table 14 and similarly found a p-value less than $10^{-7}$ in all cases. This finding is intuitively backed up by Figures 25 and 26 where the patch-based model outperformed the whole image model in nearly all of the test images. Therefore, the experiment provides statistically significant evidence that patch-based models outperform whole image models in this setting. Finally, diffusion models are a computationally expensive method that trade off reconstruction speed in exchange for higher quality images. Table 11 shows the reconstruction runtime for a single image of different methods; running the diffusion based methods over thousands of images would take many days for a single inverse problem.

---

### Official Review · Reviewer_kvAd · 2024-10-25

**Soundness:** 2
**Presentation:** 1
**Contribution:** 2
**Rating:** 5
**Confidence:** 4

**Summary:**

This paper examines the use of diffusion models in inverse problems, particularly when there’s a mismatch between the training and test data distributions (out-of-distribution, OOD). The authors investigate two settings: one where only a single measurement from an unknown test distribution is available, and another where a small sample from the test distribution is accessible. They propose a patch-based diffusion model that learns image priors from patches rather than entire images. This model includes a self-supervised loss to enhance consistency with the given measurement in the single-measurement scenario. Their experiments demonstrate that this patch-based approach produces high-quality reconstructions.

**Strengths:**

- The paper addresses an important and practical problem that commonly arises in real-world scenarios, where the generative prior distribution differs from the target distribution. This focus on out-of-distribution (OOD) issues has strong applicability in diverse settings.

- The experimental results show that the proposed patch-based model achieves better performance compared to whole-image models, underscoring the effectiveness of a patch-based approach in handling OOD challenges for inverse problems.

**Weaknesses:**

- Limited novelty. I believe that the paper’s primary contribution—a combination of a patch-based model with self-supervised loss (e.g., deep image priors) to address out-of-distribution (OOD) issues in inverse problems—builds on existing concepts. While the integration of these components to tackle a specific challenge is interesting, the novelty is somewhat limited, as each component’s effectiveness has been demonstrated in prior works. Additionally, there is a lack of theoretical justification or clear intuition for the design of Algorithm 1, which would strengthen the proposed approach.

- The experimental setup lacks clarity. Specifically, it is not explained how the authors achieve the “Whole image, correct*” model or which sampling algorithm is used. Key hypothetical models are described ambiguously, and the term “best baselines” in Table 2 is undefined, making it challenging to understand the comparisons being drawn.

- Lack of analysis. Why the proposed algorithm 1 or patch-based model is better than previous models is not clearly demonstrated in the paper.

- Missing baselines. The paper does not include comparisons with strong baseline methods for inverse problem-solving using diffusion models, such as DPS [1], DDS [2], DDNM [3], and DAVI [4]. These methods are relevant in both single-measurement and limited-sample settings. In particular, [4] addresses OOD settings, making it a highly relevant benchmark for this work.

[1] Diffusion Posterior Sampling for General Noisy Inverse Problems, ICRL23 \
[2] Decomposed Diffusion Sampler for Accelerating Large-Scale Inverse Problems, ICLR24 \
[3] Zero-Shot Image Restoration Using Denoising Diffusion Null-Space Model, ICLR23 \
[4] Diffusion Prior-Based Amortized Variational Inference for Noisy Inverse Problems, ECCV24

**Questions:**

Please provide clarity on points raised under the weaknesses section.

---

> ### Author Response · Authors · 2024-11-21
> **Response to reviewer kvAd**
>
> We thank the reviewer for their valuable feedback and insights.
>
> **Comment: Limited novelty…contribution builds on existing concepts**
>
> Response: Compared to previous patch-based model works, the main difference is that the problem setting is different. Past works studied patch-based diffusion models in a traditional generative model setting: given a dataset of images, learn the distribution of those images. In our work, we studied diffusion models in an OOD setting, where no or very limited data belonging to the test data distribution is available, so the goal is to adapt a pretrained diffusion model to a new distribution, either from a single measurement or a very small dataset. Therefore, while the networks in previous patch-based works are always trained from scratch, the networks in our work are adjusted from an existing distribution, and we show how this method drastically reduces the data requirement.
>
> Furthermore, we provide novel experimentation and analysis of how patch-based diffusion models can avoid memorization and overfitting when being adapted to OOD data. While previous works showed that using patch-based models to solve inverse problems can lead to better results in settings of limited data, they did not explicitly illustrate that patch-based models can avoid issues of memorization and overfitting that whole-image models run into. Our work tackles these issues directly through Figure 7 which illustrates that whole image models memorize the training data when fine-tuning from a very small dataset while patch-based models have greater generalizability. Figures 4 and 5 also illustrate this point: when fine-tuning the networks for a longer period of time, the quality of reconstructed images drops substantially for the whole image models while that of the patch-based models remains relatively steady.
>
> In the single measurement setting, DIP-based self-supervised models tend to overfit to the data. Therefore, past works have used LoRA to limit the expressiveness of the network and prevent overfitting. We show in Table 10 that patch-based models can avoid overfitting to the measurement and that early stopping is not necessary when refining the network at each diffusion iteration, even when LoRA is not applied.
>
> In the revised paper, we also now provide novel theoretical analysis of the proposed algorithms not present in previous works. This theoretical analysis serves two purposes. Firstly, we show how in theory, using the patch-based model allows us to perform a type of data augmentation in the single measurement setting, which reinforces the point that patch-based models help guard against overfitting. Secondly, we provide more theoretical grounding for the self-supervised network refining method used in Algorithm 1 and show that this network refining process converges.
>
> **Comment: Lack of theoretical justification of Algorithm 1**
>
> Response: We added theoretical analysis of Algorithm 1 to Section A.7.1. This provides a theoretical argument for why patch-based diffusion models should outperform whole-image models in the single measurement setting. Furthermore, we show that the network refining process used in each diffusion iteration converges.
>
> **Comment: Experimental setup lacks clarity**
>
> Response: The “whole-image, correct” model indicates training a diffusion model to learn the prior of the whole image using a large dataset of in-distribution data (either CT data or CelebA facial data). Then, since it is assumed that the network has learned the correct prior, we use a traditional diffusion inverse solving algorithm that does not involve network refining. To maintain consistency with the experiments where network refining is used, we used Langevin dynamics for the sampling algorithm with conjugate gradient descent to enforce data consistency. We added the pseudocode for the reconstruction algorithm in Algorithm 2 of Appendix A.7.2.
>
> In Table 2, “best baselines” refers to the best baseline out of the non-diffusion baselines shown in Table 1, i.e., ADMM-TV, PnP-ADMM, and PnP-RED. We did this because these experiments would have been identical for Table 1 and Table 2, so we chose to only repeat the results of the best baseline in Table 2 to avoid redundancy.

---

> ### Author Response · Authors · 2024-11-21
> **Response to reviewer kvAd (part 2)**
>
> **Comment: Missing baselines**
>
> Response: [1], [2], and [3] are all “general purpose” inverse solvers, which are all assumed to learn the prior from in-distribution training data and then conduct sampling on testing samples to obtain restored images from the trained model. Thus, these methods are not designed to solve OOD problems when the testing sample is out of the training distribution, and they do not provide ways to refine the network on the fly during testing sampling. Furthermore, [2] was applied to solve 3D image reconstruction problems while our submission focuses on 2D inverse problems. The general method that [2] uses for enforcing data consistency is conjugate gradient descent, which is the same as in our approach for both the whole-image model and patch-based model experiments. Hence, the “whole image, naive” and “patches, naive” experiments in Table 1 can be thought of as 2D versions of [2].
>
> Although these methods are not designed for OOD problems so they may not be a strong baseline in the OOD setting, to address the reviewer’s comments, we conduct additional experiments to apply these methods to test samples that are out of the training distribution, the setting used in this work. Table 16 shows the results of new comparison experiments. We also showed the visual results in Figures 29 and 30, where many artifacts are clearly visible as expected. In particular, smooth artifacts are clearly visible since the ellipse phantoms that were used to train the networks are generally smooth.
>
> [4] directly trained a network to learn the posterior distribution of the image data from the measurement given by $p(x_0|y)$. This can be seen from Algorithm 1 in [4] which requires paired data between $x_0$ and $y$ and trains the network $I_\phi$. Therefore, for different inverse problems, new networks must be retrained even if the underlying dataset is the same. On the other hand, unconditional generative methods like our proposed method (in both settings) are flexible and the same network can be used for different inverse problems. Therefore, a direct comparison between using our method for a specific inverse problem and using [4] with a network trained specifically for that inverse problem would be unfair. Moreover, Algorithm 1 in [4] requires an approximation of the score function at different timesteps $s_\psi(x_t, t)$, which in turn requires a large quantity of in-distribution training data that is not available in the single measurement setting.
>
> In the small dataset setting, we assume that, after fine-tuning, our network is sufficiently in distribution so that no refining on the fly is needed. Then we directly applied [1], [2], and [3] with the fine-tuned network and reported the results in Table 17. For all the experiments we see that DDS [2] performed the best, and hence we use a similar method of conjugate gradient descent in our main experiments.

---

### Official Review · Reviewer_t78A · 2024-10-31

**Soundness:** 3
**Presentation:** 2
**Contribution:** 2
**Rating:** 5
**Confidence:** 5

**Summary:**

This paper shows that the patch-based diffusion model can be a good solution for mismatching distribution inverse problems, compared to the conventional whole-image diffusion model. The authors study this setting where (1) only measurements are available and (2) very small ID samples are available, the results are decent.

**Strengths:**

1. The paper is easy to follow.
2. The topic of mismatching distribution inverse problems is timely and important.
3. Using patch-based diffusion models for mismatching distribution inverse problems is natural.

**Weaknesses:**

1. I doubt the contribution of the work. As it is not new to solve inverse problems with patch-based diffusion models[1], the finding of this paper 'whole-image models are prone to memorization and overfitting, while a patch-based model can resolve these issues' is already clarified in [1].

2. Lack of theoretical analysis. It is straightforward that patch-based diffusion models are suited for mismatching distribution inverse problems, as they can avoid memorization and overfitting ID data. It would be good if the authors could provide some theory on this argument.

3. Minor:

      Errors in Eq. 2 and Eq. 3;

      Better to repaint Fig. 1 instead of directly copying it from [1] without citation.


[1] Learning Image Priors through Patch-based Diffusion Models for Solving Inverse Problems. Jason Hu, et al.

**Questions:**

1. Can the authors provide some figures of training data? I have seen Fig. 20 but I still can not imagine the training data.
2. Why Fig. 6 does not have the same notations as Fig. 3? It looks really confusing.
3. I am wondering if the proposed method can scale up. Since patch-based diffusion models should aim for large images, the experiments on CelebA 256*256 are insufficient.

I am willing to raise the score if the concerns are well-addressed.

---

> ### Author Response · Authors · 2024-11-21
> **Response to reviewer t78A**
>
> **Comment: The finding of the paper is already clarified in [1]**
>
> Response: Compared to [1], the main difference is that the problem setting is different. The authors of [1] studied patch-based diffusion models in a traditional generative model setting: given a dataset of images, learn the distribution of those images. In our work, we studied diffusion models in an OOD setting, where no or very limited data belonging to the test data distribution is available, so the goal is to adapt a pretrained diffusion model to a new distribution, either from a single measurement or a very small dataset. Therefore, while the networks in [1] are always trained from scratch, the networks in our work are adjusted from an existing distribution, and we show how this method drastically reduces the data requirement.
>
> Furthermore, compared to [1], we provide novel experimentation and analysis of how patch-based diffusion models can avoid memorization and overfitting when being adapted to OOD data. While [1] showed that using patch-based models to solve inverse problems can lead to better results in settings of limited data, they did not explicitly illustrate that patch-based models can avoid issues of memorization and overfitting that whole image models run into. Our work tackles these issues directly through Figure 7 which illustrates that whole image models memorize the training data when fine-tuning from a very small dataset while patch-based models have greater generalizability. Figures 4 and 5 also illustrate this point: when fine-tuning the networks for a longer period of time, the quality of reconstructed images drops substantially for the whole image models while that of the patch-based models remains relatively steady.
>
> In the revised paper, we also now provide novel theoretical analysis of the proposed algorithms that was not present in [1]. This theoretical analysis serves two purposes. Firstly, we show how in theory, using the patch-based model allows us to perform a type of data augmentation in the single measurement setting, which reinforces the point that patch-based models help guard against overfitting. Secondly, we provide more theoretical grounding for the self-supervised network refining method used in Algorithm 1 and show that this network refining process converges.
>
> **Comment: Lack of theoretical analysis**
>
> Response: We added theoretical analysis of Algorithm 1 in Section A.7.1. This provides a theoretical argument for why patch-based diffusion models should outperform whole image models in the single measurement setting. Furthermore, we show that the network refining process used in each diffusion iteration converges.
>
> **Comment: Errors in Eq. 2 and 3 and repaint Fig. 1**
>
> Response: We fixed the errors in eq. 2 and 3 and remade Fig. 1. These changes are highlighted in blue in the revision.
>
> **Comment: Can the authors provide some figures of training data**
>
> Response: We added more examples of the phantom training data in Appendix A.7.6 in Figures 31 and 32.
>
> **Comment: Why Fig. 6 does not have same notation as Fig. 3**
>
> Response: Fig. 3 shows the results of the single measurement setting while Fig. 6 shows the results of the small dataset setting. To avoid redundancy, we did not show the reconstructions in Fig. 6 that were also shown in Fig. 3. For instance, the reconstructions done using “whole, naive”, “patch, naive”, “whole, correct”, and “patch, correct” are the same for the single measurement setting and the small dataset setting (with the first two of these being done with the network trained on the out-of-distribution training data (phantom ellipse) and the latter two using the network trained on a large quantity of in-distribution data (CT images). To further reduce redundancy we opted not to show the results of every non-diffusion baseline that we ran in Fig. 6, with the baselines being ADMM-TV, PnP-ADMM, and PnP-RED. However, we still showed the best reconstruction out of these three baselines, labeled “Best baseline” in Fig. 6. The figure still shows that our proposed method obtains the best visual results in the small dataset setting.
>
> **Comment: Can the proposed method scale up**
>
> Response: To show that our method scales to larger images, we have added experiments on 512×512 images for 60 view CT reconstruction and deblurring with a 17×17 uniform kernel. We used the AAPM dataset for the CT experiments and the FFHQ dataset for the deblurring experiments. More details and the results can be found in Table 15 and Figures 27 and 28 of section A.7.3.

---

> > ### Comment · Reviewer_t78A · 2024-11-25
> >
> > I thank the authors for the additional experiments. As promised, I have thus raised my score to 5. However, I think the contribution of the paper is limited to justify a higher score.

---

> > > ### Author Response · Authors · 2024-11-25
> > >
> > > Thank you for taking the time to read our response. If there are any other questions or issues feel free to let us know and we will do our best to address them.

---

### Official Review · Reviewer_cXcQ · 2024-11-04

**Soundness:** 3
**Presentation:** 3
**Contribution:** 2
**Rating:** 6
**Confidence:** 4

**Summary:**

The paper proposes to use patch-based diffusion models for solving inverse problems with mismatched training and test distributions. The authors address the challenge of artifacts and hallucinations in image reconstructions when the training and test datasets are not aligned. They propose a patch-based approach that leverages image patches to learn priors, demonstrating the effectiveness of the proposed method in scenarios with limited data availability.

**Strengths:**

1. The writing is clear so easy to follow.

2. The motivation of using patch-based prior for better generalizability is reasonable.

3. The proposed method addresses an important practical problem.

**Weaknesses:**

1. It is not clear why Eq. (11) can address "The image that is being reconstructed might not come from the distribution of the training images". I recommend the authors to provide more detailed discussion.

2. The proposed method is similar to Deep Diffusion Image Prior for Efficient OOD Adaptation in 3D Inverse Problems. It could be beneficial to discuss the similarity and difference to highlight the contribution of this paper.

**Questions:**

I would like to see what the results would be like when applying the method to the black whole imaging problem [1] where the true prior is unavailable.

[1] Wu, Zihui, et al. "Principled Probabilistic Imaging using Diffusion Models as Plug-and-Play Priors." arXiv preprint arXiv:2405.18782 (2024).

---

> ### Author Response · Authors · 2024-11-21
> **Response to reviewer cXcQ**
>
> We thank the reviewer for their valuable feedback and insights.
>
> **Comment: Not clear why eq. (11) can address…**
>
> Response: Taking the specific example of our work where the network has initially been trained on ellipse phantoms, and we are trying to perform CT reconstruction. Thus, the testing data distribution (CT images) for the target task is not the same as training data distribution (ellipse phantoms images). At some diffusion time step t, the quantity $D(x_t|y)$ is roughly the expectation of the clean image given the measurement. Since the prior was learned over ellipse phantoms, this expectation will likely be close to an image consisting of such phantoms. However, such an image will not be close to satisfying data consistency, i.e., $y$ will not be close to $A \cdot D(x_t|y)$. Hence, by minimizing the loss function measuring the difference between $y$ and $A \cdot D(x_t|y)$ to update the network parameters, we force the prior learned by the network to be adjusted in such a way that the resulting expected clean image is closer to satisfying data consistency.
>
> **Comment: Proposed method is similar to DDIP3D**
>
> Response: Compared to DDIP3D, we provide novel experimentation and analysis of how patch-based diffusion models can avoid memorization and overfitting when being adapted to OOD data. In particular, DDIP3D uses 2D whole-image models to solve 3D image reconstruction problems, which tends to lead to overfitting. Our work elucidates these issues directly through Figure 7 which illustrates that whole-image models memorize the training data when fine-tuning from a very small dataset while patch-based models have greater generalizability. Figures 4 and 5 also illustrate this point: when fine-tuning the networks for a longer period of time, the quality of reconstructed images drops substantially for the whole image models while that of the patch-based models remains relatively steady.
>
> In the single measurement setting, DIP-based self-supervised models such as those used in DDIP3D tend to overfit to the data. Therefore, DDIP3D used LoRA to limit the expressiveness of the network and prevent overfitting. We show in Table 10 that patch-based models can avoid overfitting to the measurement and that early stopping is not necessary when refining the network at each diffusion iteration, even when LoRA is not applied.
>
> In the revised paper, we also now provide novel theoretical analysis of the proposed algorithms that was not present in DDIP3D. This theoretical analysis serves two purposes. Firstly, we show how in theory, using the patch-based model allows us to perform a type of data augmentation in the single measurement setting, which reinforces the point that patch-based models help guard against overfitting. Secondly, we provide more theoretical grounding for the self-supervised network refining method used in Algorithm 1 and show that this network refining process converges.
>
> **Comment: What would the results be like when applying the method to the black hole imaging problem where the true prior is unavailable**
>
> Response: In our work, for the single measurement setting, we are assuming the true prior is unavailable and we are only given a measurement from an unknown test distribution. This is similar to the problem setting of [1]. The approach of [1] is different in that it examines the reconstructed image under a variety of different possible assumptions for the prior, whereas we do not make any assumption on the true prior and simply refine our pretrained network based on the measurement. After our code is released, those with more domain expertise would be able to apply our method to the black hole imaging problem of [1]. We modified the introduction of the paper to clarify this point.
>
> [1] Wu, Zihui, et al. "Principled Probabilistic Imaging using Diffusion Models as Plug-and-Play Priors." arXiv preprint arXiv:2405.18782 (2024).

---

### Official Review · Reviewer_VPj1 · 2024-11-04

**Soundness:** 3
**Presentation:** 2
**Contribution:** 2
**Rating:** 5
**Confidence:** 3

**Summary:**

The paper considers the problem of adapting diffusion models trained on a domain A  to the task of solving reconstruction problems on another domain B, and investigate the cases where a single measurement from B or a small number of samples from B are available.
To that end, the authors investigate deep diffusion image prior (DDIP) adaption methods, which were originally proposed to be used with whole-image diffusion models, and combine with the recently proposed patch-based diffusion models. The experimental results on CT and natural image datasets shows that patch-based diffusion models are more robust to fine-tuning on small dataset as well as to adapting the network weights for single-measurement domain adaption.

**Strengths:**

- The considered domain adaption tasks are important for practical application.
- The observation that patch-based diffusion models are more robust to finetuning / out-of-distribution tasks is interesting and relevant.

**Weaknesses:**

- While it is important to improve upon the considered tasks, and the paper presents good results on that task, I am unsure about the contribution of the paper. The paper combines the existing SCD/DDIP method with the recently proposed patch-based diffusion models and their inverse-problem solver (PaDIS), but it seems that this is only a matter of replacing the whole-image with patch-based diffusion models and their score calculation method. The robustness to OOD data of patch-based models is important, but the relation to the experiments on small datasets presented by [1] is unclear.
- The results in Table 1 show that the main performance gains are due to using SCD/DDIP and using that with Patch-based DMs increases the PSNR by at most 1dB. While this is certainly an improvement, using the patch-based model in-distribution already increases the PSNR.
- In section 3, the authors introduce the patch-based diffusion prior method, where they seem to have copied Figure 1 and the text (with small adaptions) from the original paper. I think this should be stated more clearly. Moreover, some parts in the argumentation seem to missing. As an example, L.198-199: "represents the aforementioned bordering region", but the bordering region has not really been mentioned before (in contrast to the original text). Restating the assumed probability distribution in L.195, but it would be helpful to also state the motivation (the calculation of the score model based on the patch scores).

References:
- [1]: Hu et al (2024): Learning Image Priors through Patch-based Diffusion Models for Solving Inverse Problems

**Questions:**

- In their appendix, [1] provide experimental results when training whole image and patch-based models on training datasets of different sizes, and similarly observe that PaDIS remains visually more consistent in contrast to the whole-image models. Could the authors elaborate how their experiments and observations relate and potentially complement those of [1]?
- See also the weaknesses above.

---

> ### Author Response · Authors · 2024-11-21
> **Response to reviewer VPj1**
>
> We thank the reviewer for their valuable feedback and insights.
>
> **Comment: I am unsure about the contribution of the paper…the robustness to OOD data is important but the relation to the experiments on small datasets presented by [1] is unclear**
>
> Response:  Compared to [1], the main difference is that the problem setting is different. The authors of [1] studied patch-based diffusion models in a traditional generative model setting: given a dataset of images, learn the distribution of those images. In our work, we studied diffusion models in an OOD setting, where no data (or very limited data) belonging to the test data distribution is available, so the goal is to adapt a pretrained diffusion model to a new distribution, either from a single measurement or a very small dataset. Therefore, while the networks in [1] are always trained from scratch, the networks in our work are adjusted from an existing distribution, and we show how this method drastically reduces the data requirement.
>
> Furthermore, compared to [1] and SCD/DDIP, we provide novel experimentation and analysis of how patch-based diffusion models can avoid memorization and overfitting when being adapted to OOD data. While [1] showed that using patch-based models to solve inverse problems can lead to better results in settings of limited data, they did not explicitly illustrate that patch-based models can avoid issues of memorization and overfitting that whole-image models run into. Our work tackles these issues directly through Figure 7 that illustrates that whole-image models memorize the training data when fine-tuning from a very small dataset while patch-based models have greater generalizability. Figures 4 and 5 also illustrate this point: when fine-tuning the networks for a longer period of time, the quality of reconstructed images drops substantially for the whole image models while that of the patch-based models remains relatively steady.
>
> In the single measurement setting, DIP-based self-supervised models such as those used in SCD/DDIP tend to overfit to the data. Therefore, SCD/DDIP used LoRA to limit the expressiveness of the network and prevent overfitting. Table 10 shows that patch-based models can avoid overfitting to the measurement and that early stopping is not necessary when refining the network at each diffusion iteration, even when LoRA is not applied.
>
> The revised paper also now provides novel theoretical analysis of the proposed algorithms that was absent from previous works. This theoretical analysis serves two purposes. Firstly, we show how in theory, using the patch-based model allows us to perform a type of data augmentation in the single measurement setting, which reinforces the point that patch-based models help guard against overfitting. Secondly, we provide more theoretical grounding for the self-supervised network refining method used in Algorithm 1 and show that this network refining process converges.
>
> **Comment: The main performance gains are due to using SCD/DDIP and using the patch-based DMs increases the PSNR by at most 1db**
>
> Response: We acknowledge that SCD/DDIP already brings a performance improvement in whole-image models. But firstly, this is expected, as the main advantage of using diffusion models for solving inverse problems is that they provide a strong prior when the measurements are compressed and/or lossy. When the prior is incorrect, we lose this advantage completely and it is unsurprising that the reconstructed image quality is poor. Thus methods such as SCD/DDIP which adjust the network are expected to yield better results. Secondly, the advantage of using patch-based models is learning a better and more robust prior. In addition, patch-based diffusion models can avoid memorization and overfitting when being adapted to OOD data, as shown in Figure 7. Figures 4 and 5 also illustrate this point: when fine-tuning the networks for a longer period of time, the quality of reconstructed images drops substantially for the whole image models while that of the patch-based models remains relatively steady. Thus, in practice, early stopping is required for whole-image models to avoid a performance drop.

---

> ### Author Response · Authors · 2024-11-21
> **Response to reviewer VPj1 (part 2)**
>
> **Comment: Using the patch-based DMs already increases the PSNR**
>
> While [1] indeed showed that the patch-based model improves performance over the whole image model, this was in settings with limited in-distribution training data in the scale of hundreds to thousands of samples, where the models are assumed to be trained from scratch. This is a different setting from our work, where we push the scale of the limited dataset to just 10 samples which makes it extremely hard to train any model from scratch. Thus, we also assume a pretrained diffusion prior is available from a different training domain where a large-scale dataset is available such as synthetic ellipsoid images. This also matches practical scenarios where a model pretrained on data-abundant domains can be utilized and transferred to help with model training in data-scarce domains.
>
> In conclusion, our work shows that the patch-based model is more readily adapted to a different test distribution either via training on a very small dataset or a single measurement. Finally, the last two rows of Table 1 show the results of using the patch-based model and the whole-image model when a large amount of in-distribution training data was available (and no network refining was done on the fly). The results are very similar for these two models, which shows that when a large-scale dataset is available, both these models are able to learn a strong prior and may achieve similar performance for inverse problem solving.
>
> **Comment: Some parts of the argumentation are missing**
>
> Response: We have added more clarifications in Section 3.1 with the changes being highlighted in blue, so that the text is more self-contained. We also added some explanations behind the motivation for using the probability model.
>
> **Comment: How do the experiments and observations relate and complement those of [1]**
>
> Response: For Table 5 of the appendix of [1], the authors trained networks from scratch using datasets that consisted of images drawn from the same distribution as the test distribution. Hence, there was no distribution mismatch in that case, and the experiment showed that the patch-based model is more readily trained using limited data than the whole-image model. In our work, for the small dataset setting, we first trained both networks using a large quantity of data from the (typically synthetic) training distribution, and then fine-tuned the networks using an extremely small dataset from a different test distribution.
>
> In addition, since the networks in [1] were trained from scratch, more data was required: the smallest datasets on which experiments were performed in [1] contained 144 images. In this work, we push the limit of the number of samples to only 10 images to fine-tune the network from a pretrained out-of-distribution prior. Consequently, due to the pretrained out-of-distribution prior and the reduction of training data, our model can converge much faster with significantly reduced training time: Figure 4 shows that we are able to fine-tune a patch-based model in only about 2 hours, while [1] required 12-24 hours to train the patch-based models. Thus, our results complement the work of [1] in the sense that: [1] shows patch-based diffusion models are easier to train from scratch in settings of limited data; this work shows that patch-based diffusion models are also easier to fine-tune from a pretrained model with few data samples. We clarified these points in section A.3 of the revised paper.

---

> > ### Comment · Reviewer_VPj1 · 2024-12-02
> >
> > Thanks to the authors for the detailed answer. So while [1] has shown that patching helps to prevent overfitting in the in-distribution setup, the authors demonstrated that it also aids fine-tuning to a small number of OOD examples or single measurements. I consider these findings interesting and important. However, similar to other reviewers, I am concerned about limited contribution, and keep my score.
> >
> > I appreciate the efforts towards a theoretical justification, and would like to give some comments in the following.
> > - The theory for CT argues that minimising $\|y - A f_{\theta}(x)\|^2$ converges. This, however, is not specific to CT or patch-based models, and does not explain the difference in performance to me (as it is suggested in the beginning of the theoretical sketch).
> > - The sketch further explains that minimizing $L(\theta) = \|y - A D_{\theta}(x)\|^2$ with patching can be understood as minimizing an upper bound $L'(\theta)$ to (an approximation of) the whole-image loss. This new loss might be more robust in the sense of data augmentation, but there could also be a gap between the losses $L(\theta)$ and $L'(\theta)$.
> > - Furthermore, it is argued that optimally one aims to reduce $L(\theta) = 0$ based on experiments performed with more update steps. However, I'd still be careful with the conclusion, as minimizing $L(\theta)$ to $0$ could might lead to overfitting to a bad solution.
> >
> > A thorough justification could greatly enhance the contribution, but requires more time for a revision I think.
> >
> > Notation:
> > - Inconsistent notation: $D_{\theta}(x, \sigma_t)$ and $D_{\theta}(x)$ are used, in the appendix $D_{\theta}(x, c)$ (with patch $c$).
> > - Typos in L.1547 (squares in the norm) and L.1573 should be $\approx$ instead of $=$ I think.

---

### Author Response · Authors · 2024-11-21
**Global response to reviewers**

We sincerely thank all the reviewers for the valuable comments and constructive feedback on our paper. We provide point-by-point responses to address each reviewer’s comments and highlight the key responses below:

*Changes to paper*

We made several corrections and additions to the paper. We added section A.7 to the appendix which features theoretical justifications of the algorithms used as well as various new experiments. We also rewrote section 3.1 and redrew Figure 1. In the main paper, all the corrections have been highlighted in blue.

*Contribution of the paper compared to previous works*

We clarify the contributions of the paper compared to previous works, especially [1] and [2]. The main difference from [1] is that the problem setting is different. The authors of [1] studied patch-based diffusion models in a traditional generative model setting: given a dataset of images, learn the distribution of those images. In our work, we studied diffusion models in an out-of-distribution (OOD) setting, where no data (or very limited data) belonging to the test data distribution is available. Thus, the goal is to adapt a pretrained diffusion model to a new distribution, either from a single measurement or a very small dataset. Whereas the networks in [1] are trained from scratch, the networks in our work are adjusted from an existing distribution. In this way, we show how this method can drastically reduce the data requirement.

Furthermore, compared to [1] and [2], we provide novel experimentation and analysis of how patch-based diffusion models can avoid memorization and overfitting when being adapted to OOD data. While [1] showed that using patch-based models to solve inverse problems can lead to better results in settings of limited data, they did not explicitly illustrate that patch-based models can avoid issues of memorization and overfitting that affect whole-image models. Our work tackles these issues directly; Figure 7 illustrates that whole-image models memorize the training data when fine-tuning from a very small dataset while patch-based models have greater generalizability. Figures 4 and 5 also illustrate this point: when fine-tuning the networks for a longer period of time, the quality of reconstructed images drops substantially for the whole-image models while that of the patch-based models remains relatively steady. Ref. [1] does not discuss fine tuning and [2] considers only whole-image models.

In the single measurement setting, DIP-based self-supervised models such as those used in [2] tend to overfit to the data. Therefore, [2] used LoRA to limit the expressiveness of the network and prevent overfitting. Our Table 10 shows that patch-based models can avoid overfitting to the measurement and that early stopping is not necessary when refining the network at each diffusion iteration, even when LoRA is not applied.

In the revised paper, we also now provide novel theoretical analysis of the proposed methods; neither [1] nor [2] have such analyses. This theoretical analysis serves two purposes. Firstly, we show how in theory, the patch-based model performs a type of data augmentation in the single measurement setting, which reinforces the point that patch-based models help guard against overfitting. Secondly, we provide more theoretical grounding for the self-supervised network refining method used in Algorithm 1 and show that this network refining process converges.

*Theoretical contribution*

We added theoretical analysis of Algorithm 1 in Section A.7.1. This provides a theoretical argument for why patch-based diffusion models should outperform whole-image models in the single measurement setting. Furthermore, we show that the network refining process used in each diffusion iteration converges.

[1]: Hu et al (2024): Learning Image Priors through Patch-based Diffusion Models for Solving Inverse Problems

[2]: Chung et al (2024): Deep Diffusion Image Prior for Efficient OOD Adaptation in 3D Inverse Problems

---

### Note · Authors · 2025-01-20

I have read and agree with the venue's withdrawal policy on behalf of myself and my co-authors.